EMBO
reports

# Proneurotrophin-3 contributes to chemotherapy-induced neuropathic pain through TrkC-mediated CCL2 elevation in DRG neurons

Dilip Sharma[1], Xiaozhou Feng[1], Bing Wang[1], Huijie Shang[1], Bushra Yasin[1], Alex Bekker[1], Huijuan Hu [ID][1,2] & Yuan-Xiang Tao [ID][1,2,3 ✉]

## Abstract

**Cancer patients undergoing treatment with antineoplastic drugs often experience chemotherapy-induced neuropathic pain (CINP), and therapeutic options for managing CINP are limited. Here, we show that systemic paclitaxel administration upregulates the expression of neurotrophin-3 (*Nt3*) mRNA and its encoded proneurotrophin-3 (proNT3) protein in the neurons of dorsal root ganglia (DRG), but not in the spinal cord. Blocking this upregulation attenuates paclitaxel-induced mechanical, heat, and cold nociceptive hypersensitivities and spontaneous pain without altering acute pain and locomotor activity in male and female mice. Conversely, mimicking paclitaxel-induced upregulation of DRG proNT3 produces enhanced responses to mechanical, heat, and cold stimuli and spontaneous pain in naive male and female mice. Mechanistically, proNT3 triggers tropomyosin receptor kinase C (TrkC) activation and participates in the paclitaxel-induced increases of C-C chemokine ligand 2 (*Ccl2*) mRNA and CCL2 protein in the DRG. Given that CCL2 is an endogenous initiator of CINP and that *Nt3* mRNA co-expresses with TrkC and *Ccl2* mRNAs in DRG neurons, proNT3 likely contributes to CINP through TrkC-mediated activation of the Ccl2 gene in DRG neurons. Thus, proNT3 may be a potential target for CINP treatment.**

**Keywords** Proneurotrophin-3; TrkC; C-C Chemokine Ligand 2; Paclitaxel-induced Peripheral Neuropathy; Dorsal Root Ganglion
**Subject Category** article

See also: D Sharma et al

## Introduction

Chemotherapy-induced neuropathic pain (CINP) is one of the side effects caused by chemotherapeutic drugs during cancer treatments (Grisold et al, 2012; Windebank and Grisold, 2008). Although the incidence and severity of CINP are correlated with the duration and dose of chemotherapy drugs, many cancer patients eventually discontinue cancer therapy due to CINP and have the reduced survival rates (Argyriou et al, 2012; Grisold et al, 2012). Current treatments of CINP are highly limited in part due to the elusive mechanisms underlying this disorder. Chemotherapy drugs have unrestricted access to the dorsal root ganglion (DRG) due to the lack of an effective vascular permeability barrier (Devor, 1999; Jimenez-Andrade et al, 2008). These drugs led to ectopic discharge, which is considered as a CINP trigger, in DRG neurons (Li et al, 2018; Li et al, 2017a; Sisignano et al, 2016; Zhang and Dougherty, 2014). The ectopic discharge may be associated with the changes in pain-related genes induced by chemotherapy drugs at both transcriptional and translational levels in DRG (Ito et al, 2017; Li et al, 2018; Li et al, 2017a; Li et al, 2014; Makker et al, 2017; Mao et al, 2019; Mao et al, 2017; Nie et al, 2018; Sisignano et al, 2016; Yilmaz and Gold, 2015; Zhang et al, 2013; Zhang and Dougherty, 2014). For example, C-C chemokine ligand 2 (CCL2, also called as MCP-1) directly excited DRG neurons through activation of its C-C motif receptor 2 (CCR2) by autocrine or paracrine processes (Jung et al, 2008; Sun et al, 2006; Van Steenwinckel et al, 2011). CCR2 activation sensitized nociceptors via transactivation of transient receptor potential channels (Jung et al, 2008). Systemic administration of the chemotherapy drugs, paclitaxel and oxaliplatin, upregulated the expression of CCL2 in the DRG neurons (Al-Mazidi et al, 2018; Curry et al, 2018; Illias et al, 2018; Wen et al, 2023; Zhang et al, 2016a). Genetic knockdown or pharmacological inhibition of DRG CCL2 alleviated these drug-induced nociceptive hypersensitivity (Al-Mazidi et al, 2018; Curry et al, 2018; Illias et al, 2018; Zhang et al, 2016a). CCL2 is likely an endogenous initiator of CINP. Understanding the mechanisms by which chemotherapy drugs cause the changes of these pain-associated genes (such as CCL2) are critical for developing a new avenue for CINP management.

Neurotrophin-3 (NT3), a member of the nerve growth factor family of neurotrophins (Richner et al, 2014), binds with high affinity to its specific receptor, tropomyosin receptor kinase C (TrkC), and less strongly to the p75 non-tyrosine receptor (NTR)

[1]Department of Anesthesiology, New Jersey Medical School, Rutgers, The State University of New Jersey, Newark, NJ 07103, USA. [2]Department of Physiology, Pharmacology & Neuroscience, New Jersey Medical School, Rutgers, The State University of New Jersey, Newark, NJ 07103, USA. [3]Department of Cell Biology & Molecular Medicine, New Jersey Medical School, Rutgers, The State University of New Jersey, Newark, NJ 07103, USA. ✉E-mail: yuanxiang.tao@njms.rutgers.edu

(Richner et al, 2014). NT3 is derived from a high-molecular-weight precursor, proneurotrophin-3 (proNT3; ~32 kDa), which is initially transcribed from *Nt3* mRNA and then cleaved by furin or other proconvertases to produce C-terminal mature NT3 protein (~14.5 kDa) (Teng et al, 2010; Yano et al, 2009). Both proNT3 and NT3 proteins are important regulators of neuronal functions, survival, growth, differentiation, cell fate choices, neurite morphology, and cell death, but their functional effects are opposite (Teng et al, 2010; Yano et al, 2009). Peripheral nerve injury increases the expression of *Nt3* mRNA in small- and medium-sized neurons of injured DRG (Kazemi et al, 2017; Wang et al, 2008). *Nt3* mRNA is also upregulated in the dorsal root, sciatic nerve, and foot skin nerves under the conditions of streptozotocin (STZ)-induced diabetic neuropathy (Cai et al, 1999). Consistently, NT3-like immunoreactivity is increased in small- and medium-sized DRG neurons following resiniferatoxin treatment (Tender et al, 2011) or removal of adjacent DRG (Wang et al, 2008) and in skin in human diabetic neuropathy (Kennedy et al, 1998). However, this immunoreactivity contains NT3 and proNT3, as the anti-NT3 antibodies used recognize both proteins. Intrathecal *Nt3* antisense oligonucleotide or local and systemic NT3 antiserum attenuated nerve injury-induced mechanical allodynia (Deng et al, 2000; Quintao et al, 2008; White, 2000; Zhou et al, 2000a). The proNT3-mediated effects in these studies cannot be excluded, given that NT3 is a C-terminal part of proNT3. One early study showed that intrathecal NT3 produced mechanical allodynia in normal rats (White, 1998). In contrast, the following works reported that intrathecal NT3 inhibited nerve injury-induced thermal hyperalgesia (Wilson-Gerwing et al, 2005) and that NT3 directly delivered into DRG did not affect mechanical nociceptive threshold (Zhou et al, 2000b). It appears that the expression of NT3 and proNT3 in the DRG and their functions in neuropathic pain, including CINP, are still elusive.

In this study, we reported the expression of proNT3, but not NT3, in the DRG. Systemic injection of the chemotherapy drug paclitaxel upregulated proNT3 in the DRG, but not in the spinal cord. More importantly, this upregulation was required and sufficient for the paclitaxel-induced CINP through TrkC-mediated CCL2 elevation in the DRG neurons.

## Results

### proNT3 expression is increased in the DRGs after systemic paclitaxel injection

To demonstrate the role of NT3 and proNT3 in CINP, we first used the anti-NT3 antibody that recognizes both proteins (Shen et al, 2013; Yano et al, 2009) and examined their abundances in two pain-related regions, DRG and spinal cord, from CD1 mice after systemic administration of paclitaxel. Consistent with our previous studies (Mao et al, 2019; Yang et al, 2021), intraperitoneal (i.p.) injection of paclitaxel, not vehicle, produced robust and long-lasting mechanical allodynia, evidenced by marked increases in paw withdrawal frequencies in response to 0.07 g and 0.4 g von Frey filaments on both left (Fig. 1A,B) and right sides. This nociceptive hypersensitivity occurred on day 7 after the first paclitaxel injection and persisted for at least 14 days (Fig. 1A,B). Correlating with these behavioral changes, the expression of *Nt3* mRNA and proNT3

protein was time-dependently upregulated in the DRGs after paclitaxel injection (Figs. 1C,D and EV1A–C). Unexpectedly, NT3 was undetected in naive DRG and the paclitaxel-treated DRGs (Fig. EV1A–C). The amount of *Nt3 mRNA* in the L3/4 DRGs was increased by 1.4-fold ($P > 0.05$), 1.8-fold ($P < 0.05$), 2.1-fold ($P < 0.01$), and 1.8-fold ($P < 0.05$) on days 7, 10, 14, and 21, respectively, after the first paclitaxel injection as compared to the corresponding vehicle-treated mice (Fig. 1C). Consistently, the level of proNT3 protein in the L3/4 DRGs was elevated by 1.4-fold ($P < 0.05$), 1.5-fold ($P < 0.05$), 2.2-fold ($P < 0.01$), and 1.4-fold ($P < 0.05$) on days 7, 10, 14, and 21 after the first paclitaxel injection, respectively, as compared to the corresponding vehicle-treated mice (0 day; Fig. 1D). As expected, i.p. injection of vehicle did not change basal levels of *Nt3 mRNA* and proNT3 protein in the L3/4 DRGs during the observation period (Fig. 1C,D). The amount of proNT3 protein was not altered in the L3/4 spinal cord during the observation period after the first paclitaxel injection (Fig. 1E).

The distribution pattern of proNT3 in the DRGs of CD1 mice was examined after systemic injection of paclitaxel. *Nt3* mRNA was expressed in the cellular cytoplasm and co-localized with β-tubulin III (a specific neuronal marker), but not with glutamine synthetase (GS, a marker for satellite glial cells), in DRG cells of paclitaxel- or vehicle-injected mice (Fig. 2A,B), indicating that *Nt3* mRNA expresses exclusively in DRG neurons. Approximately 5.5% and 29.6% of β-tubulin III-labeled DRG neurons were positive for *Nt3* mRNA in the vehicle and paclitaxel mice, respectively, on day 14 after the first paclitaxel or vehicle injection (Fig. 2A). A cross-sectional area analysis of neuronal somata showed that about 25% of *Nt3* mRNA-labeled neurons were small ($< 500\ \mu m^2$ in area), 36% for medium ($500–1000\ \mu m^2$ in area), and 39% for large ($>1000\ \mu m^2$ in area) in the DRGs of the paclitaxel-treated mice (Fig. 2C). Consistently, about 30% of *Nt3* mRNA-labeled neurons were positive for CGRP (a marker for small DRG peptidergic neurons; Fig. 2D,H), 34% for NF200 (a marker for medium/large neurons and myelinated Aβ fibers; Fig. 2E,H), 33% for IB4 (a marker for small nonpeptidergic neurons; Fig. 2F,H) and 18% for tyrosine hydroxylase (TH, a marker for small low-threshold neurons; Fig. 2G,H) in the paclitaxel-treated DRGs.

Taken together, the neuronal distribution pattern of *Nt3* mRNA and paclitaxel-induced increases of *Nt3* mRNA and pro-NT3 protein in the DRGs suggests proNT3 involvement in paclitaxel-induced CINP.

### Effect of blocking DRG proNT3 upregulation on paclitaxel-induced nociceptive hypersensitivities

To examine whether DRG proNT3 upregulation participated in the paclitaxel-induced CINP, we first blocked DRG proNT3 upregulation through single microinjection of *Nt3* short interfering RNA (NT3 siRNA; dissolved in PBS) into unilateral L3/4 DRGs of adult CD1 male mice 7 days after the first i.p. injection of vehicle or paclitaxel. Negative control scrambled siRNA (NC siRNA; dissolved in PBS) was used as a control. As expected, the level of proNT3 was increased by 1.7-fold ($P < 0.01$) in the microinjected L3/4 DRGs from paclitaxel plus NC siRNA-treated group compared to that in vehicle plus NC siRNA-treated group, 14 days after the first paclitaxel injection (Fig. 3A). This increase was not seen in the paclitaxel plus NT3 siRNA-treated group (Fig. 3A).

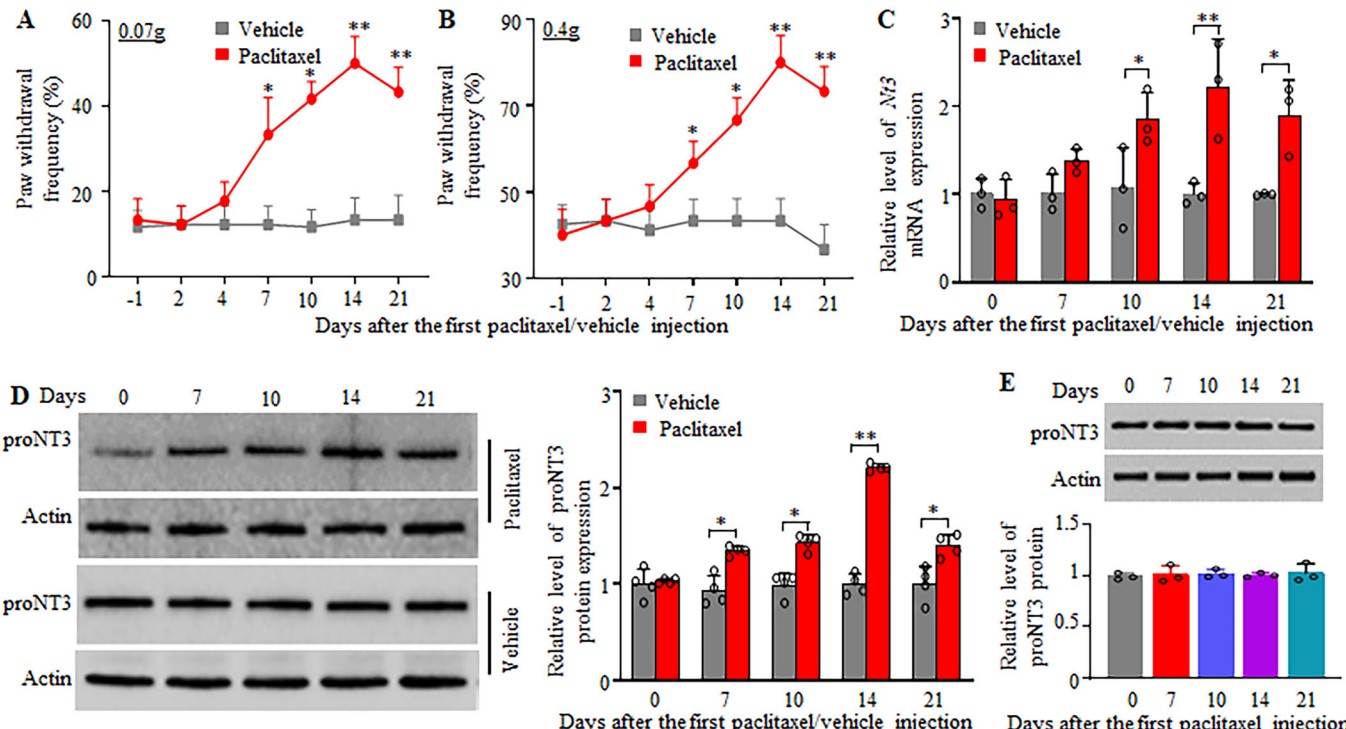

**Figure 1. proNT3 abundance is increased in the DRG after paclitaxel injection.**

Data: mean ± SD. (A, B) Intraperitoneal injection of paclitaxel produced mechanical allodynia as evidenced by the increases in paw withdrawal frequencies in response to 0.07 g (A) and 0.4 g (B) von Frey filaments on the left side. n = 12 mice/time point. *$P < 0.05$, **$P < 0.01$ versus the vehicle-treated group at the corresponding time points by two-way ANOVA with repeated measures followed by post hoc Tukey test. (C, D) The levels of *Nt3* mRNA (C) and proNT3 protein (D) in L3/4 DRGs on different days as indicated after the first injection of paclitaxel or vehicle. Representative Western blots (left) and a summary of the densitometric analysis (right) are shown. n = 3 biological repeats (3 mice)/time point. *$P < 0.05$, **$P < 0.01$ by two-way ANOVA followed by post hoc Tukey test. (E) The amount of proNT3 protein in the L3/4 spinal cord on different days as indicated after the first injection of paclitaxel. n = 3 biological repeated (3 mice)/time point. $P > 0.05$ by one-way ANOVA followed by post hoc Tukey test. Source data are available online for this figure.

Basal levels of proNT3 were not significantly changed in the microinjected L3/4 DRGs of the NT3 siRNA plus vehicle-treated group (Fig. 3A). Paclitaxel injection produced mechanical allodynia in response to mechanical stimuli (0.07 g and 0.4 g von Frey filaments) and heat and cold hyperalgesia in response to heat and cold stimuli starting day 7 after the first paclitaxel injection and persisting for at least 7 days in NC siRNA-microinjected mice (Fig. 3B–H). These nociceptive hypersensitivities were significantly alleviated on the ipsilateral (not contralateral) side in the paclitaxel plus NT3 siRNA-treated group on days 10, 12, and 14 post the first paclitaxel injection (Fig. 3B–H). Neither NT3 siRNA microinjection nor NC siRNA microinjection altered basal paw withdrawal responses on either side in the vehicle-treated group (Fig. 3B–H).

Neuronal hyperexcitability in the DRGs caused by systemic administration of paclitaxel triggers the hyperactivation of neurons and astrocytes in the spinal cord dorsal horn likely through augmenting the release of neurotransmitters and/or neuromodulators in primary afferents (Wen et al, 2023; Yang et al, 2021). To further confirm our behavioral observations above, we examined whether DRG microinjection of NT3 siRNA also affected the hyperactivities of spinal cord dorsal horn neurons and astrocytes, as documented by increases in the levels of phosphorylated extracellular signal-regulated kinase ½ (p-ERK1/2, a marker for neuronal hyperactivation) and glial fibrillary acidic protein (GFAP,

a marker for astrocyte hyperactivation), respectively. The levels of p-ERK1, p-ERK2 and GFAP were increased by 3.0 ($P < 0.01$)-, 3.1 ($P < 0.01$)- and 2.1 ($P < 0.05$)-fold, respectively, in the ipsilateral L3/4 dorsal horn of the paclitaxel plus NC siRNA-treated group, as compared to those in vehicle plus NC siRNA-treated group on day 14 post-the first paclitaxel or vehicle injection (Fig. 3I). These increases were not observed in the paclitaxel plus NT3 siRNA-treated mice (Fig. 3I). No significant differences in basal amounts of total ERK1/2 were seen in the ipsilateral L3/4 dorsal horn among all treated groups (Fig. 3I). Neither siRNA affected basal levels of p-ERK1/2, ERK1/2 and GFAP in the ipsilateral L3/4 dorsal horn in the vehicle-treated group (Fig. 3I).

Similar findings were observed in the paclitaxel or vehicle-treated female mice with DRG microinjection of NT3 siRNA or NC siRNA (Fig. 4A–I).

To exclude the possibilities that the observed behavioral changes above may be caused by siRNA off-targets and/or microinjection-induced tissue damage, we carried out i.p. injection of paclitaxel or vehicle 7 days before i.p. injection of tamoxifen (1 mg/mouse daily for 7 consecutive days; dissolved in 90% of sunflower seed oil and 10% of dimethyl sulfoxide) in male NT3[f/f] mice and NT3 cKD mice. As expected, the levels of *Nt3* mRNA and proNT3 protein were increased by 2.7- and 2.5-fold, respectively, in the bilateral L3/4 DRGs from the paclitaxel plus tamoxifen-treated NT3[f/f] mice, as

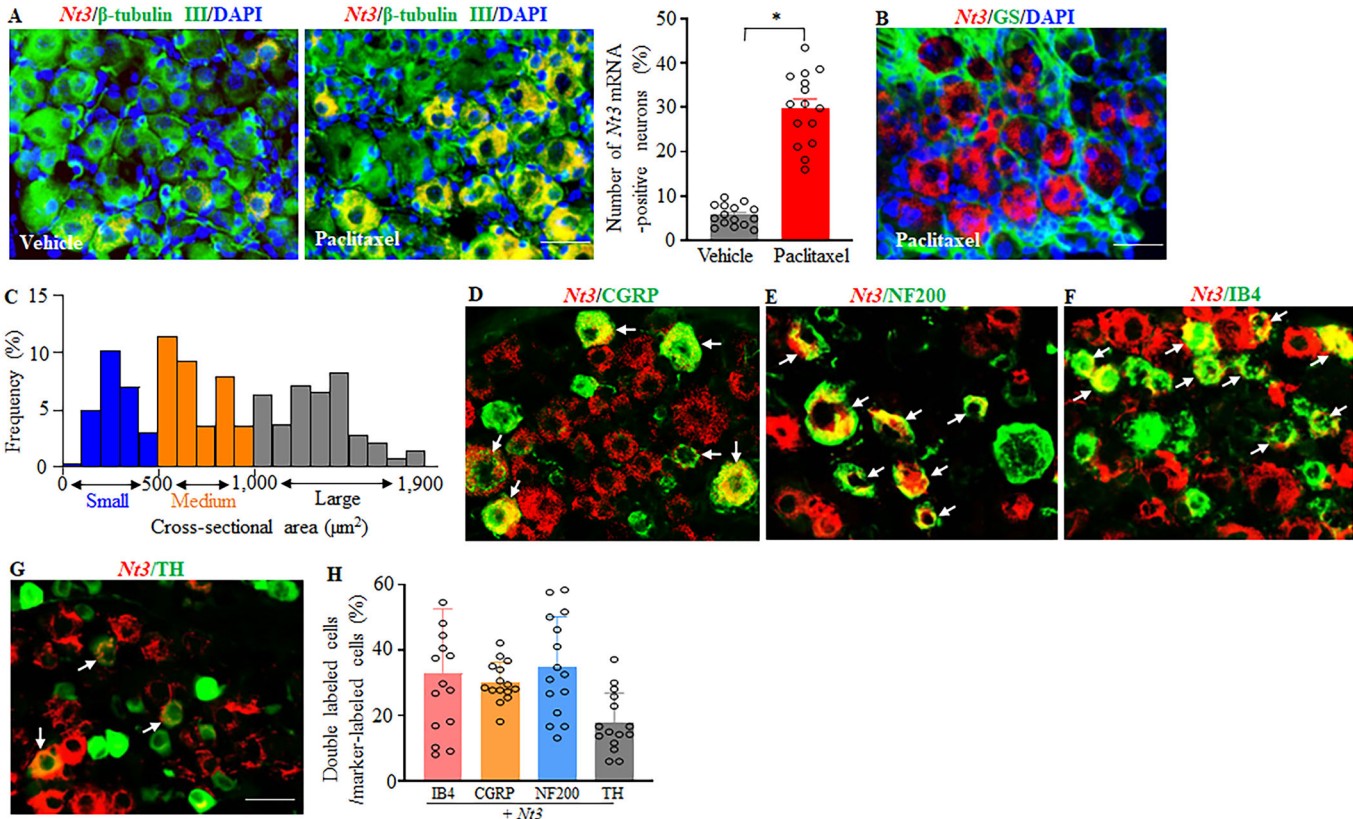

**Figure 2.  Distribution of Nt3 mRNA in the DRG from the vehicle- or paclitaxel-treated mice.**

Data: mean ± SD. (A, B) Representative images of in situ hybridization histochemistry (ISHH) for *Nt3* mRNA (red) and immunohistochemistry for β-tubulin III (green; **A**) or glutamine synthetase (GS; green; **B**) in the L3/4 DRGs 14 days after the first paclitaxel or vehicle injection. Cellular nuclei were labeled by 4′, 6-diamidino-2-phenylindole (DAPI; blue). $n = 9$–12 sections from three mice/group. *$P < 0.05$ by two-tailed unpaired Student $t$ test. Scale bars: 50 µm. (**C**) Size distribution of *Nt3* mRNA-labeled neuronal somata in the L3/4 DRGs 14 days after the first paclitaxel injection. Small, 25%; medium, 36%; large, 39%. (**D–H**) Representative images of ISHH for *Nt3* mRNA (red) and immunohistochemistry for different DRG cell markers (calcitonin gene-related peptide (CGRP; green; **D**), neurofilament 200 (NF200; green; **E**), isolectin B4 (IB4; green; **F**), or tyrosine hydroxylase (TH; green; **G**)) in the L3/4 DRGs 14 days after the first paclitaxel injection. Arrows: double-labeling neurons. Statistical summary of number of double-labeling neurons (**H**). $n = 9$–12 sections from 3 mice. Scale bar: 50 µm. Source data are available online for this figure.

compared to those in the vehicle plus tamoxifen-treated NT3[f/f] mice on day 28 post the first paclitaxel injection (Fig. 5A,B). These increases were significantly blocked in the paclitaxel plus tamoxifen-treated NT3 cKD mice (Fig. 5A,B). Basal levels of *Nt3* mRNA and proNT3 protein were not altered in the bilateral L3/4 DRGs from the vehicle plus tamoxifen-treated NT3 cKD mice (Fig. 5A,B). Blocking the paclitaxel-induced increase of proNT3 in the bilateral L3/4 DRGs of male NT3 cKD mice following tamoxifen injection impaired the paclitaxel-induced mechanical allodynia and heat and cold hyperalgesia on both left and right sides from day 12 to day 28 post the first paclitaxel injection (Fig. 5C–I). These NT3 cKD mice with tamoxifen injection failed to exhibit significant preference toward either the saline- or lidocaine-paired chamber on day 21 post the first paclitaxel injection, indicating the reduction in the paclitaxel-induced stimulation-independent spontaneous ongoing pain (Fig. 5J,K). As expected, the paclitaxel-evoked increases of neuronal/astrocyte hyperactivities in the L3/4 dorsal horn from the tamoxifen-injected NT3[f/f] mice were not detected in the tamoxifen-injected NT3 cKD mice on day 28 following the first paclitaxel injection (Fig. 5L). After vehicle

injection, basal paw withdrawal responses on the bilateral sides of the tamoxifen-treated NT3[f/f] mice and NT3 cKD mice were not changed during the observation periods (Fig. 5C–I). All siRNA-microinjected male and female mice and tamoxifen-treated NT3[f/f] male mice or NT3 cKD male mice exhibited normal locomotor activity (Table EV1).

## Mimicking the paclitaxel-induced DRG proNT3 upregulation produces nociceptive hypersensitivities in naive mice

We next inquired whether the paclitaxel-induced DRG proNT3 upregulation was sufficient for CINP. To this end, we mimicked the paclitaxel-induced DRG proNT3 increase through microinjection of AAV5 expressing full-length *Nt3* mRNA (AAV5-NT3) into unilateral L3/4 DRGs of naive male CD1 mice. AAV5 expressing green fluorescent protein (AAV5-GFP) was used as a control. As anticipated, the levels of *Nt3* mRNA and proNT3 protein in the ipsilateral L3/4 DRGs were increased by 3.4- and 2.2-fold, respectively, in mice microinjected with AAV5-NT3 compared to

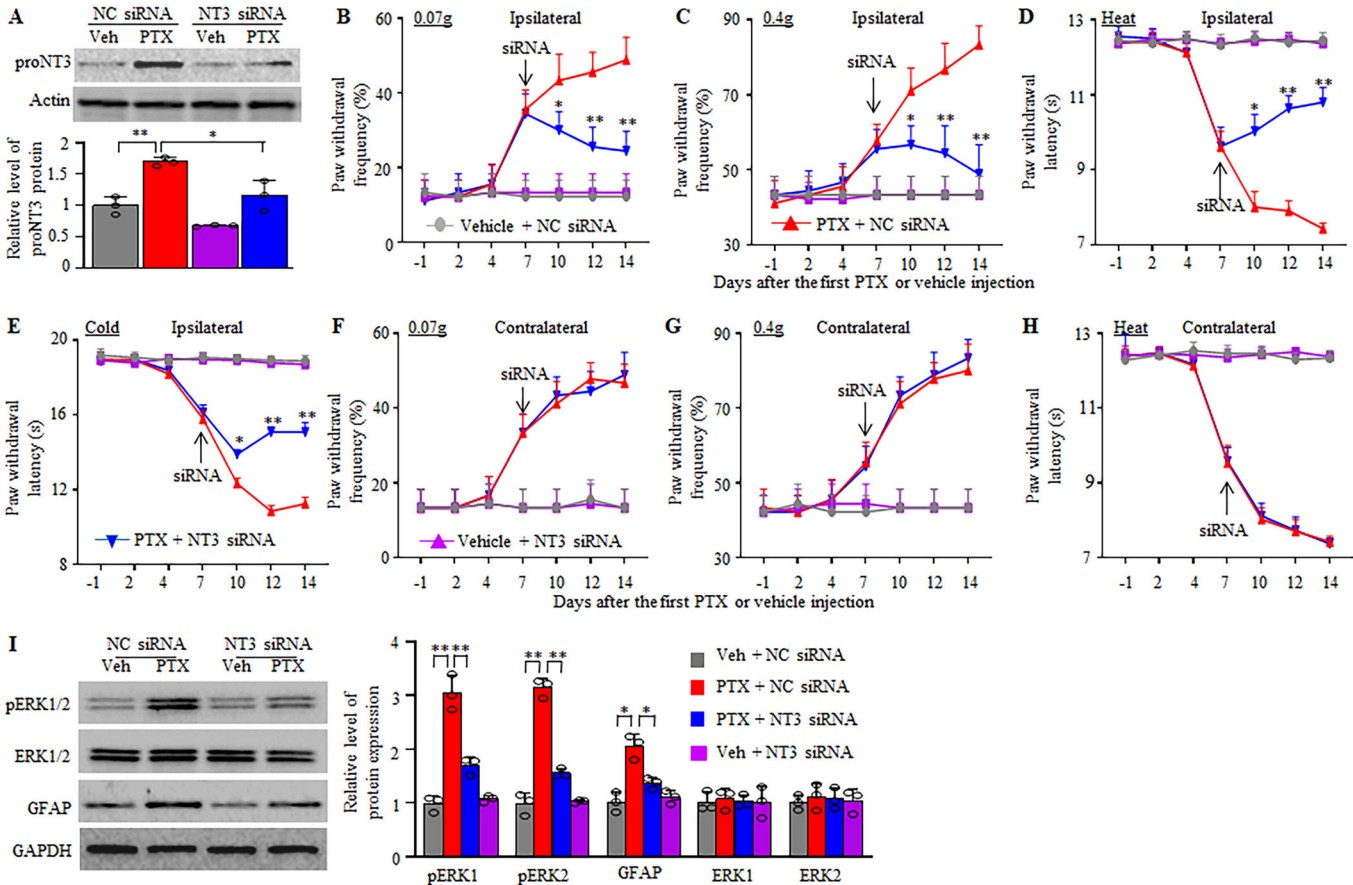

**Figure 3. Effect of microinjection of NT3 siRNA or negative control scrambled siRNA (NC siRNA) into unilateral L3/4 DRGs on paclitaxel-induced nociceptive hypersensitivity in male mice.**

Data: mean ± SD. (A) Expression of proNT3 protein in the ipsilateral L3/4 DRGs 14 days after the first injection of paclitaxel (PTX) or vehicle (Veh) from the NT3 siRNA- or NC siRNA-microinjected male mice. $n = 3$ biological repeats (6 mice)/group. *$P < 0.05$, **$P < 0.01$ by two-way ANOVA followed by post hoc Tukey test. (B–H) Paw withdrawal frequency to 0.07 g (B, F) and 0.4 g (C, G) von Frey filament stimuli and paw withdrawal latency to heat (D, H) and cold (E) stimuli on the ipsilateral (B–E) and contralateral (F–H) side on days as indicated from male mice with DRG microinjection of NT3 siRNA or NC siRNA 7 days after the first injection of paclitaxel (PTX) or vehicle. $n = 9$ mice/group. *$P < 0.05$, **$P < 0.01$ versus the paclitaxel plus NC siRNA-treated group at the corresponding time points by three-way ANOVA with repeated measures followed by post hoc Tukey test. (I) Levels of p-ERK1/2, total ERK1/2, and GFAP in the ipsilateral L3/4 dorsal horn 14 days after the first injection of paclitaxel (PTX) or vehicle (Veh) from male mice with DRG microinjection of NT3 siRNA or NC siRNA. $n = 3$ mice/group. *$P < 0.05$, **$P < 0.01$ by two-way ANOVA followed by Tukey post hoc test. Source data are available online for this figure.

those microinjected with AAV5-GFP 8 weeks after viral micro-injection (Fig. 6A,B). More interestingly, DRG microinjection of AAV5-NT3, but not AAV5-GFP, produced significant increases in paw withdrawal frequencies in response to 0.07 g and 0.4 g von Frey filament stimuli and decreases in paw withdrawal latencies in response to heat and cold stimuli on the ipsilateral (not contralateral) side (Fig. 6C–F). These nociceptive hypersensi-tivities developed 4 weeks post-viral microinjection and persisted for at least 4 weeks (Fig. 6C–F). This microinjection also led to stimulation-independent spontaneous pain and increased neuronal and astrocyte activities in the ipsilateral L3/4 dorsal horn 8 weeks after viral microinjection (Fig. 6G–I). Neither viral microinjection affected locomotor functions (Table EV1). Similar observations were found in naive female mice after DRG microinjection of AAV5-NT3 or AAV5-GFP (Fig. 7A–I; Table EV1). Together, these findings indicate that mimicking the paclitaxel-induced DRG proNT3 increase produces both

spontaneous and evoked nociceptive hypersensitivities in naive male and female mice.

## TrkC, but not p75 receptor, mediates the proNT3-induced nociceptive hypersensitivity

We further determined how DRG upregulated proNT3 caused nociceptive hypersensitivities. The biological effects of proNT3 are produced via proNT3 binding to TrkC and p75NTR (Richner et al, 2014). We examined whether DRG TrkC and p75NTR knockdown through microinjection of specific TrkC siRNA and p75NTR siRNA, respectively, into the ipsilateral L3/4 DRGs 35 days after AAV5-NT3 microinjection into unilateral L3/4 DRGs on nocic-eptive hypersensitivities in naive CD1 male mice. As expected, basal levels of *Nt3* mRNA and proNT3 protein were significantly increased in the ipsilateral L3/4 DRGs from the AAV5-NT3 (not AAV5-GFP) plus control scrambled siRNA-, TrkC siRNA- or

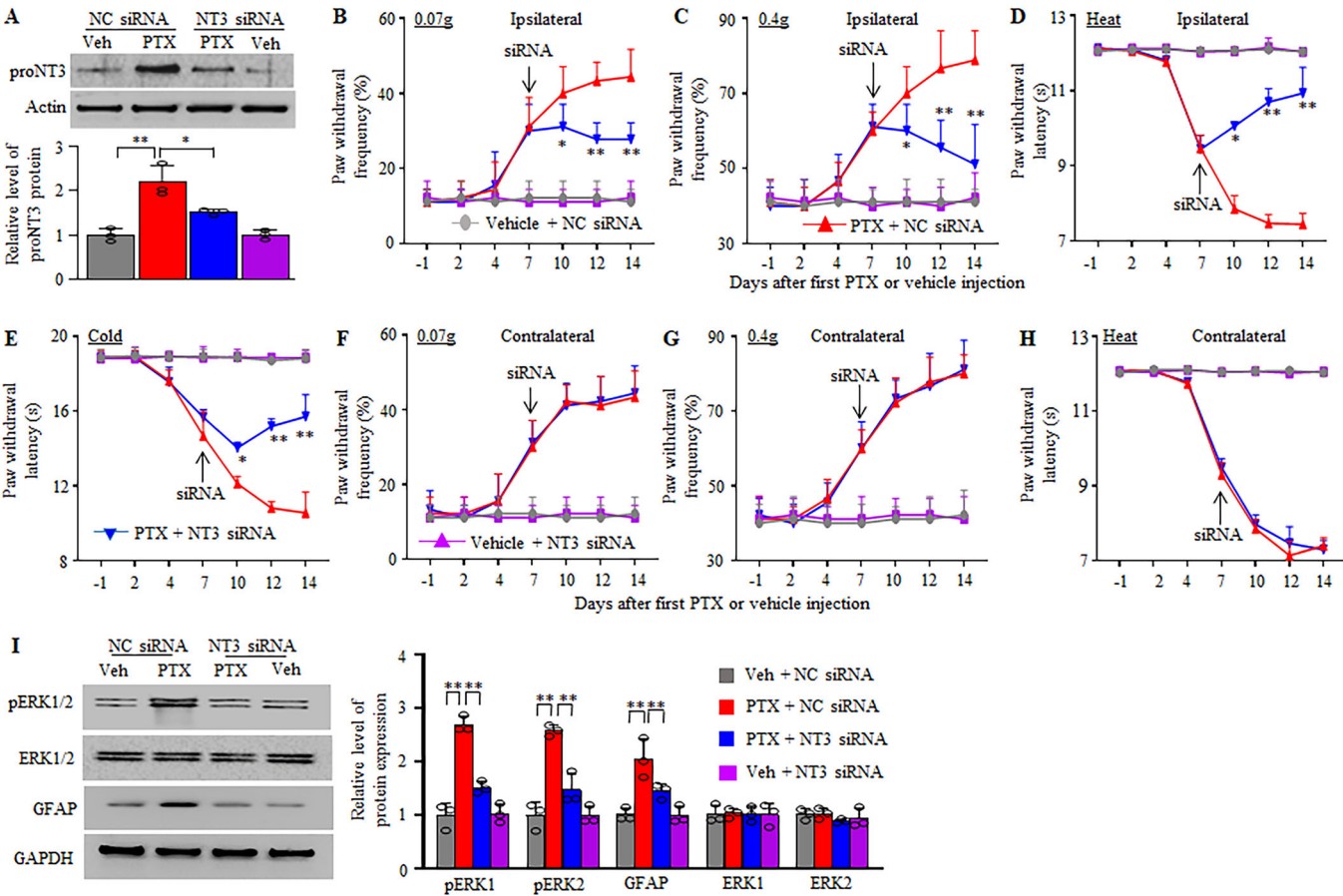

**Figure 4. Effect of microinjection of NT3 siRNA or negative control scrambled siRNA (NC siRNA) into unilateral L3/4 DRGs on paclitaxel-induced nociceptive hypersensitivity in female mice.**

Data: mean ± SD. (A) Expression of proNT3 protein in the ipsilateral L3/4 DRGs 14 days after the first injection of paclitaxel (PTX) or vehicle (Veh) from the NT3 siRNA- or NC siRNA-microinjected female mice. $n = 3$ biological repeats (6 mice)/group. *$P < 0.05$, **$P < 0.01$ by two-way ANOVA followed by post hoc Tukey test. (B–H) Paw withdrawal frequency to 0.07 g (B, F) and 0.4 g (C, G) von Frey filament stimuli and paw withdrawal latency to heat (D, H) and cold (E) stimuli on the ipsilateral (B–E) and contralateral (F–H) side on the days as indicated from female mice with DRG microinjection of NT3 siRNA or NC siRNA on day 7 after the first injection of paclitaxel or vehicle. $n = 9$ mice/group. *$P < 0.05$, **$P < 0.01$ versus the paclitaxel plus NC siRNA-treated group at the corresponding time points by three-way ANOVA with repeated measures followed by post hoc Tukey test. (I) Levels of p-ERK1/2, total ERK1/2, and GFAP in the ipsilateral L3/4 dorsal horn 14 days after the first injection of paclitaxel (PTX) or vehicle (Veh) from female mice with DRG microinjection of NT3 siRNA or NC siRNA. $n = 3$ mice/group. **$P < 0.01$ by two-way ANOVA followed by Tukey post hoc test. Source data are available online for this figure.

p75NTR siRNA-treated groups (Fig. 8A,B), whereas basal levels of *TrkC* mRNA/protein and *p75 NTR* mRNA/protein were markedly decreased in the ipsilateral L3/4 DRGs from the TrkC siRNA plus AAV5-GFP- or AAV5-NT3-treated groups and from the p75NTR siRNA plus AAV5-GFP or AAV5-NT3, respectively, on day 42 after AAV5 microinjection (Fig. 8A,B). More importantly, the increases in paw withdrawal frequencies to 0.07 g and 0.4 g von Frey filament stimuli and decreases in paw withdrawal latencies to heat and cold stimuli on the ipsilateral side in the AAV5-NT3 plus control scrambled siRNA-treated group were impaired in the AAV5-NT3 plus TrkC siRNA-treated group, but not in the AAV5-NT3 plus p75NTR siRNA-treated group (Fig. 8C–F). None of these siRNAs altered basal responses to mechanical, heat, and cold stimuli on the contralateral side of the AAV5-NT3-treated group and on both sides of the AAV5-GFP-treated group (Fig. 8C–I). In addition, significant increases in the levels of p-ERK1/2 and GFAP in the ipsilateral L3/4 dorsal horn on day 42 post-DRG AAV5-NT3

microinjection in the control scrambled siRNA-treated group were found in the p75NTR siRNA-treated group, but not in the TrkC siRNA-treated group (Fig. 8J). No alterations in basal amounts of total ERK1/2 in the ipsilateral L3/4 dorsal horn (Fig. 8J) and locomotor functions (Table EV1) were seen among siRNA-treated groups. These findings suggest that proNT3 produces nociceptive hypersensitivity primarily through the activation of TrkC, but not p75NTR, in the DRG neurons.

## Upregulated proNT3 is responsible for the TrkC-mediated CCL2 increase in the paclitaxel-exposed DRG

Finally, we elucidated the mechanism by which DRG upregulated proNT3 contributed to the paclitaxel-induced CINP. The chemokine CCL2 (also called MCP-1) plays a pivotal role in CINP (Al-Mazidi et al, 2018; Curry et al, 2018; Illias et al, 2018; Wen et al, 2023; Zhang et al, 2016b). Given that CCL2 was a potential

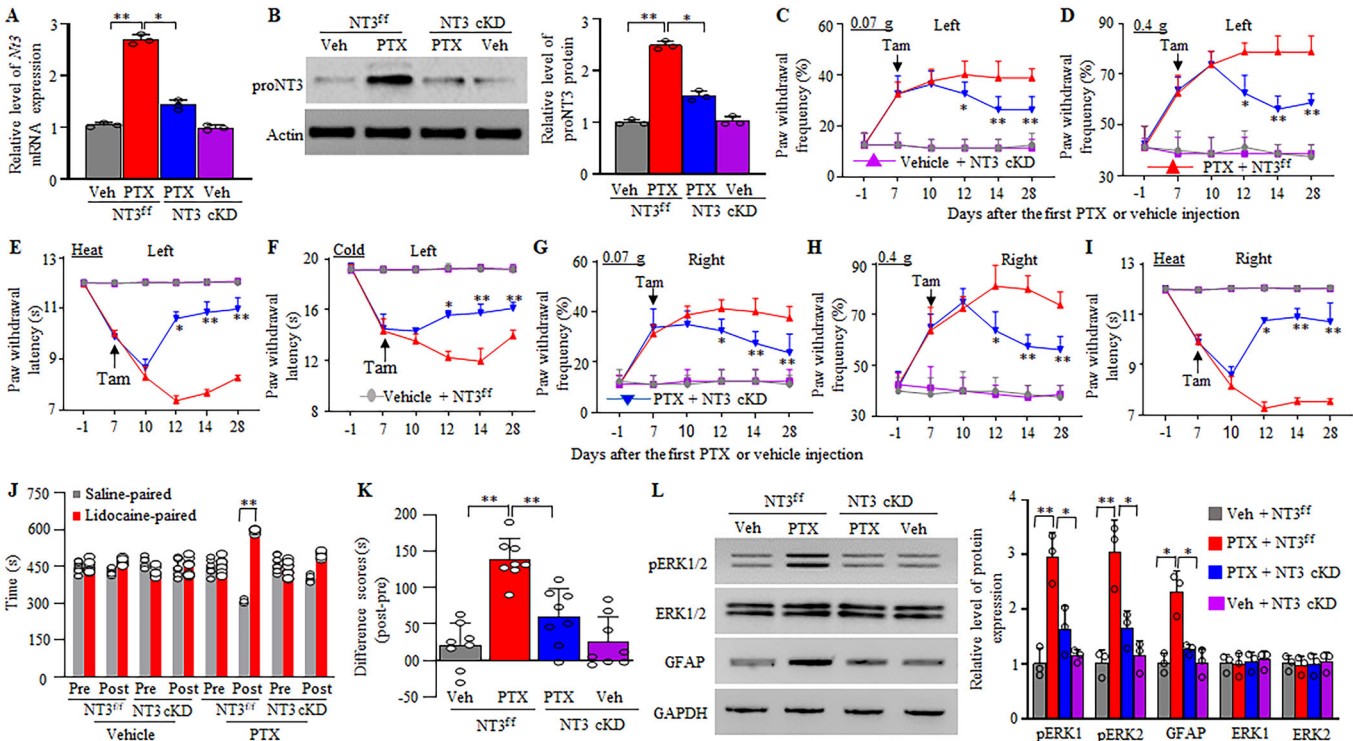

**Figure 5. Effect of genetic knockdown of *Nt3* in DRG sensory neurons on paclitaxel-induced nociceptive hypersensitivity in male mice.**

Data: mean ± SD. (A, B) Levels of *Nt3* mRNA (A) and proNT3 protein (B) in the L3/4 DRGs 28 days after the first intraperitoneal (i.p.) injection of paclitaxel (PTX) or vehicle (Veh) from the tamoxifen-treated NT3[f/f] mice and NT3 cKD mice. $n = 3$ biological repeated (3 mice)/group. *$P < 0.05$, **$P < 0.01$ by two-way ANOVA followed by post hoc Tukey test. (C–I) Paw withdrawal frequency to 0.07 g (C, G) and 0.4 g (D, H) von Frey filament stimuli and paw withdrawal latency to heat (E, I) and cold (F) stimuli on the left (C–F) and right (G–I) side on days as indicated from NT3[f/f] mice and NT3 cKD with i.p. injection of tamoxifen (Tam) daily for 7 consecutive days starting at day 7 after the first i.p. injection of paclitaxel (PTX) or vehicle. $n = 9$ mice/group. *$P < 0.05$, **$P < 0.01$ versus the paclitaxel-treated NT3[f/f] mice at the corresponding days by three-way ANOVA with repeated measures followed by post hoc Tukey test. (J, K) Spontaneous ongoing pain as assessed by the CPP test 21 days after the first i.p. injection of paclitaxel (PTX) or vehicle in the tamoxifen-treated NT3[f/f] mice and NT3 cKD mice. Time spent in each chamber (J) and difference scores for chamber preferences calculated by subtracting preconditioning (Pre) preference time from postconditioning (Post) time spent in the lidocaine-paired chamber (K). $n = 9$ mice/group. **$P < 0.01$ by three-way ANOVA with repeated measures followed by post hoc Tukey test (J) or by two-way ANOVA followed by Tukey post hoc test (K). (L) Levels of p-ERK1/2, total ERK1/2 and GFAP in the L3/4 dorsal horn 28 days after the first i.p. injection of paclitaxel (PTX) or vehicle in the tamoxifen-treated NT3[f/f] mice and NT3 cKD mice. $n = 3$ mice/group. *$P < 0.05$, **$P < 0.01$ by two-way ANOVA followed by post hoc Tukey test. Source data are available online for this figure.

downstream target of proNT3/TrkC (Salvador et al, 2014), we predicted that CCL2 might mediate the role of DRG upregulated proNT3 in the paclitaxel-induced CINP. Consistent with previous reports (Al-Mazidi et al, 2018; Curry et al, 2018; Illias et al, 2018; Wen et al, 2023; Zhang et al, 2016b), the level of CCL2 was substantially increased in the bilateral L3/4 DRGs on day 14 after the first paclitaxel injection from the NC siRNA-treated male (Fig. 9A) and female (Fig. 9B) CD1 mice or on day 28 after the first paclitaxel injection from the tamoxifen-treated NT3[f/f] male mice (Fig. 9C). These increases were significantly blocked in the paclitaxel plus NT3 siRNA-treated CD1 male (Fig. 9A) and female (Fig. 9B) mice or in the paclitaxel plus tamoxifen-treated NT3 cKD male mice (Fig. 9C). There was also a significant increase in level of TrkC in the ipsilateral L3/4 DRG on day 14 after the first paclitaxel injection from the NC siRNA-treated male (Fig. 9A) and female (Fig. 9B) CD1 mice or on day 28 after the first paclitaxel injection from the tamoxifen-treated NT3[f/f] male mice (Fig. 9C). These increases were still seen in the paclitaxel plus NT3 siRNA-treated CD1 male (Fig. 9A) and female (Fig. 9B) mice or in the paclitaxel

plus tamoxifen-treated NT3 cKD male mice (Fig. 9C). Basal levels of CCL2 and TrkC in the bilateral L3/4 DRGs from the vehicle plus NT3 siRNA-treated CD1 male and female mice or from the vehicle plus tamoxifen-treated NT3 cKD male mice were not altered (Fig. 9A–C). Consistently, microinjection of AAV5-NT3, but not AAV5-GFP, into the unilateral L3/4 DRGs markedly elevated the amount of CCL2 in the ipsilateral L3/4 DRGs 8 weeks after AAV5 microinjection in the control scrambled siRNA-treated mice (Fig. 9D). This elevation was prevented in the AAV5-NT3 plus TrkC siRNA-treated mice, although DRG microinjection of TrkC siRNA did not change the basal level of CCL2 in the AAV5-GFP-microinjected mice (Fig. 9D). To further confirm our in vivo observations above, we explored the paclitaxel on the cultured DRG neurons. The levels of *Nt3* mRNA, *Ccl2* mRNA, and their corresponding coding proteins were markedly increased in the paclitaxel plus negative control scrambled siRNA-treated DRG neurons (Fig. 9E,F). These increases were noticeably diminished in the paclitaxel plus NT3 siRNA-treated DRG neurons (Fig. 9E,F). NT3 siRNA did not alter basal levels of *Nt3* mRNA, *Ccl2* mRNA,

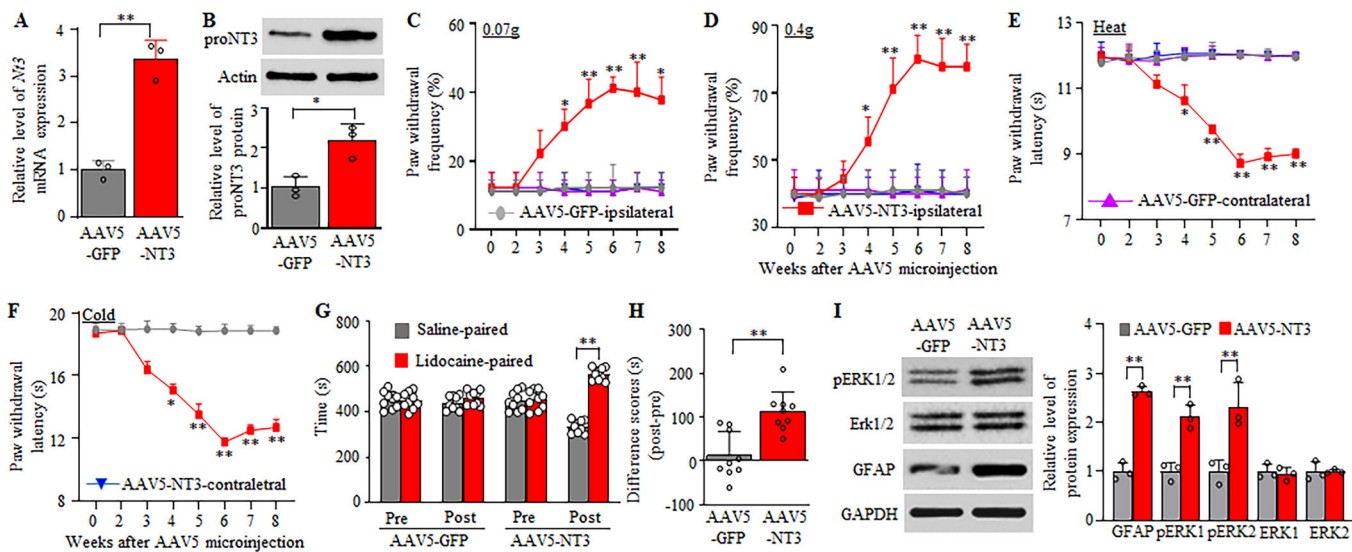

**Figure 6. Effect of DRG proNT3 overexpression on nociceptive thresholds in naive male mice.**

Data: mean ± SD. (A, B) Level of *Nt3* mRNA (A) and proNT3 protein (B) in the ipsilateral L3/4 DRGs 8 weeks after microinjection of AAV5-NT3 or control AAV5-GFP in male mice. $n = 3$ biological repeats (6 mice)/group. **$P < 0.01$ by two-tailed unpaired Student $t$ test. (C–F) Paw withdrawal frequency to 0.07 g (C) and 0.4 g (D) von Frey filament stimuli and paw withdrawal latency to heat (E) and cold (F) stimuli on the ipsilateral and contralateral sides on weeks as indicated after microinjection of AAV5-NT3 or AAV5-GFP into unilateral L3/4 DRGs. $n = 9$ mice/group. *$P < 0.05$, **$P < 0.01$ versus the AAV5-GFP group on the ipsilateral side at the corresponding time points by three-way ANOVA with repeated measures followed by Tukey post hoc test. (G, H) Spontaneous ongoing pain as assessed by the CPP test 8 weeks after microinjection of AAV5-NT3 or AAV5-GFP into unilateral L3/4 DRGs. Time spent in each chamber (G) and difference scores for chamber preferences calculated by subtracting preconditioning (Pre) preference time from postconditioning (Post) time spent in the lidocaine-paired chamber (H). $n = 9$ mice/group. **$P < 0.01$ by two-way ANOVA with repeated measures followed by post hoc Tukey test (G) or by two-tailed unpaired Student $t$ test (H). (I) Levels of p-ERK1/2, total ERK1/2, and GFAP in the ipsilateral L3/4 dorsal horn 8 weeks after microinjection of AAV5-NT3 or AAV5-GFP into unilateral L3/4 DRGs. $n = 3$ mice/group. **$P < 0.01$ by two-tailed unpaired Student $t$ test. Source data are available online for this figure.

proNT3 protein and CCL2 protein in the vehicle-treated DRG neurons (Fig. 9E,F). Similar observations were seen in human neuroblastoma cell lines exposed to the paclitaxel or vehicle and treated with NT3 siRNA or negative control scrambled siRNA (Fig. 9G,H). In addition, the transduction of AAV5-NT3, but not AAV5-GFP, into the cultured DRG neurons dramatically increased the levels of not only *Nt3* mRNA and proNT3 protein but also *Ccl2* mRNA and CCL2 protein in the negative control scrambled siRNA-treated DRG neurons (Fig. 9I,J). NT3 siRNA treatment obviously blocked these increases in the AAV5-NT3-transduced DRG neurons (Fig. 9I,J). Single-cell RT-PCR assay showed that *Nt3* mRNA co-expressed with *TrkC* mRNA and *Ccl2* mRNA in some individual large, medium, and small DRG neurons (Fig. 10A–C). Our findings suggest that upregulated proNT3 participates in paclitaxel-induced CINP at least in part due to the TrkC-mediated activation of *Ccl2* gene expression in DRG neurons.

# Discussion

Systemic injection of paclitaxel produces nociceptive hypersensitivities in mice, mimicking CINP caused by chemotherapy drugs in clinical settings. Due to the limited analgesic efficacy in CINP treatments, understanding how paclitaxel injection leads to nociceptive hypersensitivity may open new avenues for managing this disorder. The present study showed the time-dependent upregulation of *Nt3* mRNA and proNT3 protein in DRG neurons after systemic paclitaxel injection. Preventing this upregulation

blocked the paclitaxel-induced increase in DRG CCL2 expression and nociceptive hypersensitivity. Mimicking this upregulation elevated TrkC-mediated DRG CCL2 expression and enhanced TrkC-mediated responses to mechanical, heat, and cold stimuli in mice without paclitaxel injection. Given that *Nt3* mRNA co-expressed with *TrkC* mRNA and *Ccl2* mRNA in individual DRG neurons, our findings suggest that proNT3 contributes to CINP likely through TrkC-mediated activation of *Ccl2* gene in the DRG.

NT3, like brain-derived neurotrophic factor (Huang et al, 2021) and nerve growth factor (Nakahashi et al, 2014), is derived from a larger precursor, proNT3. The latter is initially transcribed from *Nt3* mRNA and then undergoes proteolytic cleavage to produce the C-terminal mature NT3 form (Teng et al, 2010; Yano et al, 2009). Thus, mature NT3 is identical to the C-terminal of proNT3. *Nt3* mRNA is upregulated in small and medium DRG neurons following peripheral nerve trauma (Kazemi et al, 2017; Wang et al, 2008) and in the dorsal root, sciatic nerve, and foot skin nerves after STZ injection (Cai et al, 1999). Consistently, NT3-like immunoreactivity was also increased in small- and medium DRG neurons following resiniferatoxin treatment (Tender et al, 2011) and removal of adjacent DRG (Wang et al, 2008). In contrast, two early studies revealed the decreases of *Nt3* and *TrkC* mRNAs and NT3 protein in skeletal muscles, DRG, or sciatic nerve of streptozotocin (STZ)-treated rats (Fernyhough et al, 1998; Ihara et al, 1996). The opposing data from these two studies might be related to performing insensitive and non-quantitative Northern blotting and ELISA assays and tissue collection at the late stage (12–14 w) of STZ rats (at this stage, peripheral nerves are

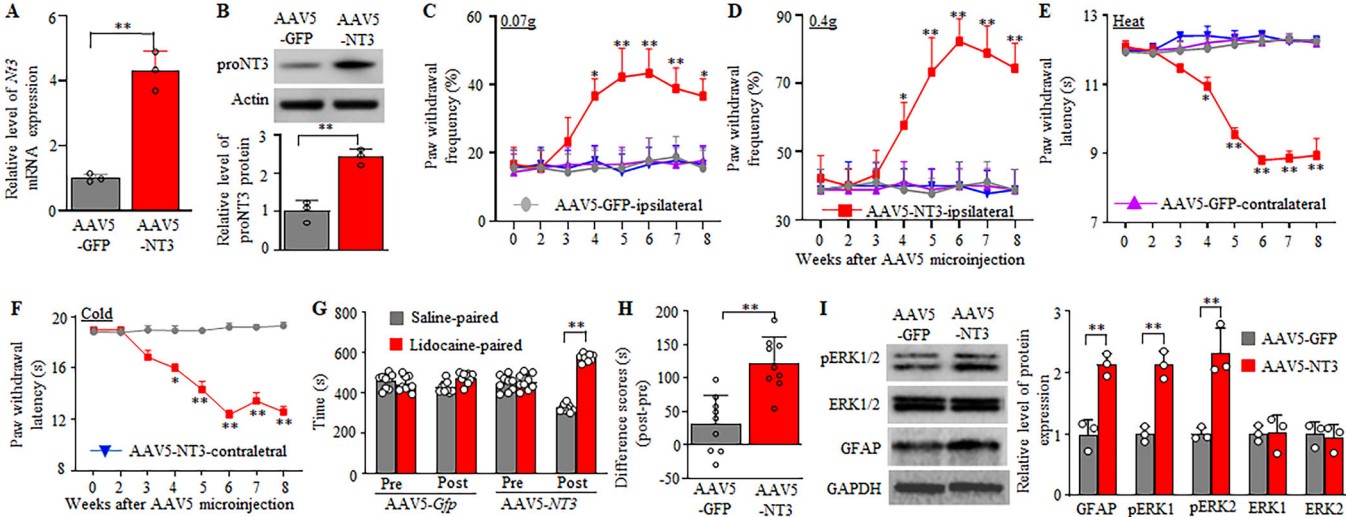

**Figure 7. Effect of DRG proNT3 overexpression on nociceptive thresholds in naive female mice.**

Data: mean ± SD. (A, B) Level of *Nt3* mRNA (A) and proNT3 protein (B) in the ipsilateral L3/4 DRGs 8 weeks after microinjection of AAV5-NT3 or control AAV5-GFP in female mice. n = 3 biological repeats (6 mice)/group. **P < 0.01 by two-tailed unpaired Student t test. (C–F) Paw withdrawal frequency to 0.07 g (C) and 0.4 g (D) von Frey filament stimuli and paw withdrawal latency to heat (E) and cold (F) stimuli on the ipsilateral and contralateral sides in weeks as indicated after microinjection of AAV5-NT3 or AAV5-GFP into unilateral L3/4 DRGs. n = 9 mice/group. *P < 0.05, **P < 0.01 versus the AAV5-GFP group on the ipsilateral side at the corresponding time points by three-way ANOVA with repeated measures followed by Tukey post hoc test. (G, H) Spontaneous ongoing pain as assessed by the CPP test 8 weeks after microinjection of AAV5-NT3 or AAV5-GFP into unilateral L3/4 DRGs. Time spent in each chamber (G) and difference scores for chamber preferences calculated by subtracting preconditioning (Pre) preference time from postconditioning (Post) time spent in the lidocaine-paired chamber (H). n = 9 mice/group. **P < 0.01 by two-way ANOVA with repeated measures followed by post hoc Tukey test (J) or by two-tailed unpaired Student t test (H). (I) Levels of p-ERK1/2, total ERK1/2, and GFAP in the ipsilateral L3/4 dorsal horn 8 weeks after microinjection of AAV5-NT3 or AAV5-GFP into unilateral L3/4 DRGs. n = 3 mice/group. **P < 0.01 by two-tailed unpaired Student t test. Source data are available online for this figure.

degraded), as one of these two studies showed an increasing tendency of DRG *TrkC* mRNA 6 weeks post-STZ (Fernyhough et al, 1998). These reported NT3 protein results cannot exclude proNT3 component, because the anti-NT3 antibody used in these studies should recognize both NT3 and proNT3. The present study used the anti-NT3 antibody against both NT3 and proNT3 in the Western blot assay and reported that the expression of proNT3, but not NT3, was detected in the DRG of naive mice (based on their molecular weights) and time-dependently increased in the DRG following systemic injection of paclitaxel. It is intriguing why proNT3 is not cleaved into mature NT3 in the DRG under normal conditions and up to 21 days post-paclitaxel injection. Whether furin and other proconvertases are absent or present at low levels in the DRG remains to be determined. The paclitaxel-induced increases in *Nt3* mRNA and proNT3 protein in the DRG, but not in the spinal cord, may be related to the fact that paclitaxel cannot penetrate through blood–brain barrier. Interestingly, an earlier increase in proNT3 protein rather than *Nt3* mRNA in the DRG on day 7 after the first paclitaxel injection was unexpectedly observed. The reason why this occurs is unclear, but it appears that paclitaxel injection may have more strongly promoted effect on proNT3 protein translation than on *Nt3* mRNA transcription and post-transcriptional modification in the DRG at an early stage. The paclitaxel-induced increase in DRG proNT3 is correlated with paclitaxel-induced nociceptive hypersensitivity. Because commercially available NT3 antibodies do not apply for immunohisto-chemistry staining, we conducted ISHH assay for detecting *Nt3* mRNA cellular distribution. We found that paclitaxel-induced *Nt3*

mRNA upregulation occurred exclusively in small, medium, and large DRG neurons. These findings suggest that upregulated proNT3 in the DRG neurons may be implicated in the paclitaxel-induced CINP. Our findings also indicate that *Nt3* gene is transcriptionally activated under the conditions of paclitaxel exposure. Although the detailed mechanisms underlying this activation are still elusive in the present study, possible involvement of transcription factors, epigenetic modifications and/or increase in RNA stability in paclitaxel-induced DRG *Nt3* mRNA upregulation cannot be ruled out, and further investigation is needed in our going research.

In this study, we employed both NT3 siRNA and sensory neuron-specific inducible NT3 cKD transgenic mice to examine the role of DRG proNT3 in the paclitaxel-induced CINP. We demonstrated that either the microinjection of NT3 siRNA into the unilateral L3/4 DRGs or i.p. injection of tamoxifen in the NT3 cKD mice blocked the paclitaxel-induced DRG proNT3 upregulation and nociceptive hypersensitivity without affecting basal (acute) pain and locomotor functions. These findings indicate the specificity and selectivity of the effects of both strategies. NT3 siRNA directly knocks down proNT3 expression in the unilateral L3/4 DRGs, but siRNA may have potential off-targets and lack the cell type transfection selectivity. In contrast, tamoxifen injection-induced proNT3 knockdown in NT3 cKD mice occurs exclusively in sensory neurons, although this knockdown is seen in all DRGs and trigeminal ganglions. Thus, the data from these two strategies complement each other. Unexpectedly, DRG microinjection of NT3 siRNA or tamoxifen injection in NT3 cKD mice failed

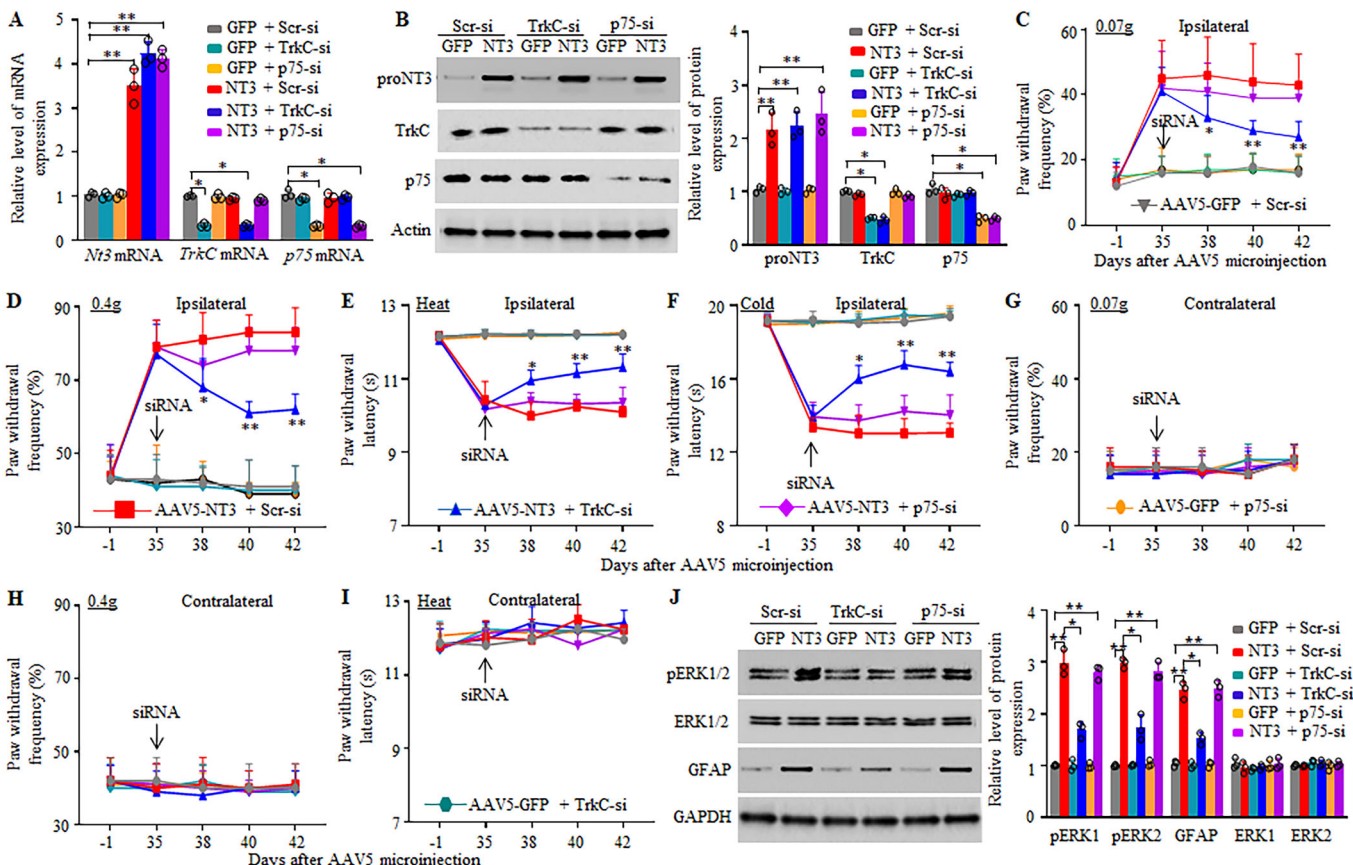

**Figure 8. Effect of DRG TrkC or p75 NTR knockdown on DRG proNT3 overexpression-induced nociceptive hypersensitivity in naive male mice.**

Data: mean ± SD. (**A, B**) Levels of *Nt3*, *TrkC*, and *p75 NTR* mRNAs (**A**) and proNT3, TrkC, and p75 proteins (**B**) in the ipsilateral L3/4 DRGs 42 days after microinjection of AAV5-NT3 (NT3) or AAV5-GFP (GFP) into unilateral L3/4 DRGs of the mice with co-microinjection of control scrambled siRNA (Scr-si), TrkC siRNA (TrkC-si), or p75 NTR siRNA (p75-si). n = 3 biological repeats (6 mice)/group. *P < 0.05, **P < 0.01 by one-way ANOVA followed by post hoc Tukey test. (**C–I**) Paw withdrawal frequencies to 0.07 g (**C, G**) and 0.4 g (**D, H**) von Frey filament stimuli and paw withdrawal latencies to heat (**E, I**) and cold stimuli (**F**) on the ipsilateral (**C–F**) and contralateral (**G–I**) sides on days as indicated after AAV-NT3 or AAV-GFP microinjection in the mice with co-microinjection of control scrambled siRNA (Scr-si), TrkC siRNA (TrkC-si), or p75 NTR siRNA (p75-si) starting at 35 days after AAV5 microinjection. n = 9 mice/group. *P < 0.05, **P < 0.01 versus the AAV5-NT3 plus control scrambled siRNA-treated group at the corresponding days by two-way ANOVA with repeated measures followed by Tukey post hoc test. (**J**) Levels of p-ERK1/2, total ERK1/2, and GFAP in the ipsilateral L3/4 dorsal horn on day 42 after microinjection of AAV5-NT3 (NT3) or AAV5-GFP (GFP) into unilateral L3/4 DRGs of the mice with co-microinjection of control scrambled siRNA (Scr-si), TrkC siRNA (TrkC-si), or p75 NTR siRNA (p75-si). n = 3 mice/group. *P < 0.05, **P < 0.01 by one-way ANOVA followed by Tukey post hoc test. Source data are available online for this figure.

to significantly knock down basal expression of *Nt3* mRNA and proNT3 protein in the DRG from the vehicle-treated mice or in the vehicle-treated cultured DRG neurons. The reasons why these two strategies did not affect basal proNT3 expression are unclear, but it is very likely that a lower basal level of proNT3 expression under normal conditions cannot be markedly further knocked down by NT3 siRNA or tamoxifen at the present dosage used.

The upregulated DRG proNT3 contributes to the paclitaxel-induced CINP likely through the TrkC-mediated increase of CCL2 in DRG neurons. Systemic administration of paclitaxel or oxaliplatin increased the expression of DRG CCL2 (Al-Mazidi et al, 2018; Curry et al, 2018; Illias et al, 2018; Wen et al, 2023; Wen et al, 2020; Zhang et al, 2016a). We also showed a robust increase in the level of CCL2 in DRGs on days 14 and 28 after the first paclitaxel injection. Pharmacological blockade or genetic knock-down of DRG CCL2 attenuated the paclitaxel-induced nociceptive hypersensitivity (Al-Mazidi et al, 2018; Curry et al, 2018; Illias et al,

2018; Zhang et al, 2016a). Increased DRG CCL2 likely is an endogenous instigator of CINP. This study demonstrated that proNT3 regulated CCL2 expression via TrkC in DRG neurons after paclitaxel injection. DRG microinjection of NT3 siRNA or tamoxifen injection in NT3 cKD mice not only attenuated the paclitaxel-induced nociceptive hypersensitivity but also blocked the paclitaxel-induced increase in DRG CCL2 in both male and female mice. Conversely, DRG overexpression of proNT3 increased basal expression of DRG CCL2 and led to both stimulation-dependent and independent nociceptive hypersensitivity in both male and female mice. The increase in DRG CCL2 induced by DRG proNT3 overexpression could be inhibited by DRG TrkC knockdown. Our in vitro experiments further revealed that CCL2 expression at both mRNA and protein levels was increased in cultured DRG neurons exposed to paclitaxel or *Nt3* mRNA overexpression. These increases were attenuated after proNT3 knockdown. Given the co-expression of *Nt3* mRNA, *TrkC* mRNA, and *Ccl2* mRNA in some individual

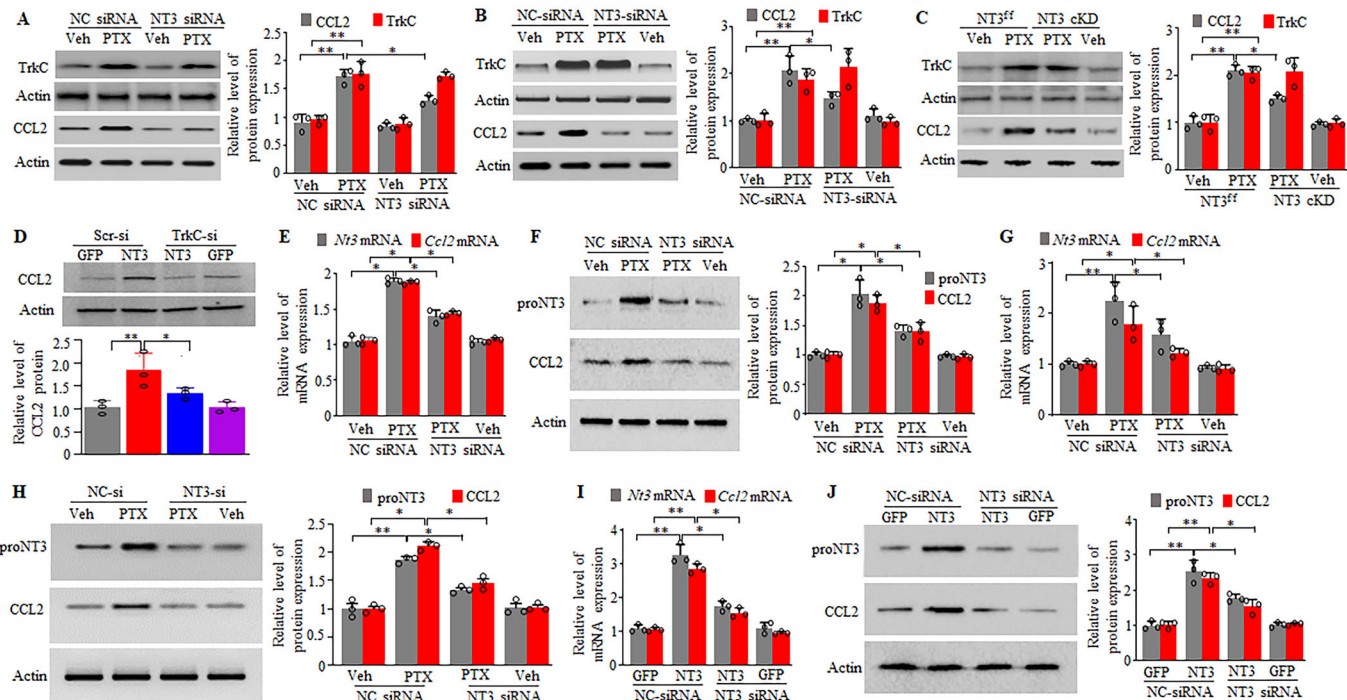

**Figure 9. Role of upregulated DRG proNT3 in paclitaxel-induced CCL2 expression in DRG.**

Data: mean ± SD. Data: mean ± SD. (A–C) Levels of CCL2 and TrkC proteins in the ipsilateral L3/4 DRGs 14 days after the first injection of paclitaxel (PTX) or vehicle (Veh) from the NT3 siRNA- or negative control scrambled siRNA (NC siRNA)-microinjected male (A) and female (B) mice or 28 days after the first injection of paclitaxel or vehicle from the tamoxifen-treated NT3$^{f/f}$ mice and NT3 cKD mice (C). $n = 3$ biological repeats (6 mice in (A, B); 3 mice in C)/group. *$P < 0.05$, **$P < 0.01$ by two-way ANOVA followed by post hoc Tukey test. (D) Level of CCL2 protein in the ipsilateral L3/4 DRGs 42 days after microinjection of AAV5-NT3 (NT3) or AAV5-GFP (GFP) into unilateral L3/4 DRGs of the mice with co-microinjection of control scrambled siRNA (Scr-si), or TrkC siRNA (TrkC-si). $n = 3$ biological repeats (6 mice)/group. *$P < 0.05$, **$P < 0.01$ by one-way ANOVA followed by post hoc Tukey test. (E–H) Levels of Nt3 mRNA (E, G), Ccl2 mRNA (E, G), proNT3 protein (F, H) and CCL2 protein (F, H) in the cultured DRG neurons (E, F) or in human neuroblastoma cell lines (G, H) with transfection of NT3 siRNA or negative control scrambled siRNA (NC siRNA) plus with exposure to paclitaxel (PTX) or vehicle (Veh). $n = 3$ biological repeats. *$P < 0.05$, **$P < 0.01$ by one-way ANOVA followed by post hoc Tukey test. (I, J) Levels of Nt3 mRNA (I), Ccl2 mRNA (I), proNT3 protein (J), and CCL2 protein (J) in the cultured DRG neurons with transduction of AAV5-NT3 (NT3) or AAV5-GFP (GFP) plus with transfection of NT3 siRNA or negative control scrambled siRNA (NC siRNA). $n = 3$ biological repeats. *$P < 0.05$, **$P < 0.01$ by one-way ANOVA followed by post hoc Tukey test. Source data are available online for this figure.

DRG neurons, the antinociception resulting from blocking the paclitaxel-induced DRG proNT3 upregulation likely stems from the failure of TrkC-mediated transcriptional activation of *Ccl2* gene in DRG neurons, although the detailed mechanisms of how proNT3/TrkC transcriptionally activates the promoter of *Ccl2* gene are still unknown. Silence of CCL2 protein expression reduced the excitability of DRG neurons (Jung et al, 2008; Sun et al, 2006; Van Steenwinckel et al, 2011), subsequently decreased the release of primary afferent transmitters/neuromodulators (Baba et al, 2003; Gao et al, 2009), and finally impaired spinal cord central sensitization formation. Indeed, we found that blocking DRG proNT3 upregulation reduced the paclitaxel-induced hyperactivation in dorsal horn astrocytes and neurons. It should be noted that proNT3 may active also as a ligand of TrkA, TrkB, and/or sortilin receptor (Bartkowska et al, 2010). The receptors-mediated role of proNT3 in CINP remains to be further studied.

In conclusion, we have explored a novel mechanism by which upregulated proNT3 increased the TrkC-mediated expression of CCL2 in DRG neurons under the conditions of paclitaxel-induced CINP. Since blocking the upregulated DRG proNT3 alleviated paclitaxel-induced CINP without altering basal or acute pain or locomotor functions, proNT3 may be a potential target for treating

this disorder. In particular, proNT3 knockdown blocked the paclitaxel-induced CCL2 expression in in vitro human neuroblastoma cell lines. This suggests that targeting proNT3 has strong potential for translation in clinical CINP management. Nevertheless, it should be noted that proNT3 is expressed in other body tissues in addition to the DRG, and potential side effects caused by targeting proNT3 should be considered.

# Methods

## Animal preparations

Adult male and female CD1 mice (2–3 months) weighing between 25 and 30 g were purchased from Charles River Laboratory (Wilmington, MA). The NT3$^{f/f}$ mice and Advillin$^{Cre-ERT2/+}$ mice (background: C57/BL6) were purchased from Jackson Laboratory (Bar Harbor, ME). To obtain the sensory neuron-specific inducible conditional proNT3 knockdown (NT3 cKD) mice, we crossbred male Advillin$^{Cre-ERT2/+}$ mice with female NT3$^{f/f}$ mice. All mice were housed in the central housing facility at Rutgers New Jersey Medical School under a standard 12-h light/dark cycle, with water and food pellets available ad libitum. The Animal Care and Use

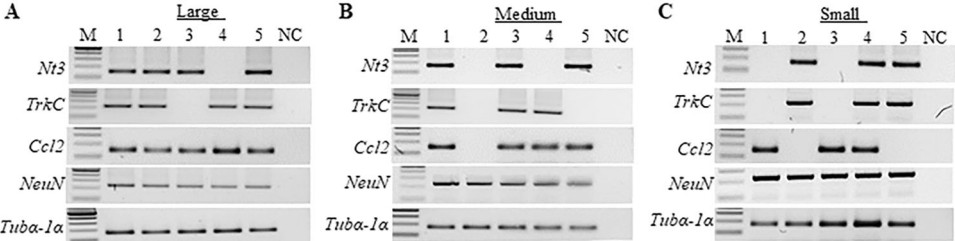

**Figure 10. Nt3 mRNA, TrkC mRNA and Ccl2 mRNA co-exist in individual DRG neurons.**

Co-expression analysis of *Nt3* mRNA, *TrkC* mRNA and *Ccl2* mRNA in individual large (>35 μm in diameter; (**A**), medium (25–35 μm in diameter; (**B**), and small (<25 μm in diameter; (**C**) DRG neurons from naive mice (n = 2) by single-cell RT-PCR assay. *NeuN* mRNA is used as a neuronal marker. *Tuba-1a* mRNA is used as a loading control. M: DNA ladder marker. NC: no cDNA. n = 5 neurons/size. Source data are available online for this figure.

Committee at Rutgers New Jersey Medical School approved all experimental procedures (Protocol number: PROTO201702655), which were also consistent with the ethical guidelines of the US National Institutes of Health and the International Association for the Study of Pain. The animals were randomly assigned to different groups. The experimenters were blinded to all treated conditions. All efforts were made to minimize mouse suffering and to reduce the number of mice used.

## Paclitaxel CINP mouse model

The mouse model of paclitaxel CINP was conducted as described previously (Jia et al, 2022; Mao et al, 2019; Wei et al, 2021; Wen et al, 2023; Yang et al, 2021). Briefly, paclitaxel (6 mg/ml in 50% Cremophor El (Sigma-Aldrich, St. Louis, MO) and 50% ethanol (Sigma-Aldrich)) was diluted in sterile saline to a final concentration of 0.4 mg/ml and administered intraperitoneally at a dose of 4 mg/kg every other day for 8 consecutive days. The vehicle was prepared in the same way as preparation described above without paclitaxel.

## DRG microinjection

DRG microinjection was conducted as described with minor modification (Jia et al, 2022; Mao et al, 2019; Wei et al, 2021; Wen et al, 2023; Yang et al, 2021). After the mouse was anesthetized with isoflurane, a dorsal midline incision was made in the lower lumbar back region. The unilateral L4 and/or L3 articular processes were exposed and then removed. Viral solution (1 μl/DRG; 4–9 × 10^12 VG/ml; dissolved in PBS) or siRNA solution (1 μl/DRG; 40–80 μM; Santa Cruz Biotechnology (catalog number: sc-42126 for NT3 siRNA)) was injected into ipsilateral exposed L4 or L3/4 DRGs with the use of a glass micropipette connected to a Hamilton syringe. siRNA was dissolved in TurboFect in vivo transfection reagent (Thermo Scientific Inc, Pittsburgh, PA) to improve delivery and prevent degradation of siRNA (Li et al, 2017b). After injection, the 10-min pipette retention was used before it was removed. The surgical field was irrigated with sterile saline, and the skin incision closed with wound clips. None of the microinjected mice showed signs of paresis or other abnormalities.

## Behavioral testing

The evoked behavioral testing, including mechanical, heat, and cold tests, was carried out in sequential order at 1-hour intervals.

Conditional place preference (CPP) testing was performed 6 weeks after microinjection. Locomotor function testing was carried out before the tissue collection.

Paw withdrawal thresholds in response to mechanical stimulation were measured with two calibrated von Frey filaments (0.07 and/or 0.4 g, Stoelting Co., Wood Dale, IL) as described (Jia et al, 2022; Mao et al, 2019; Wei et al, 2021; Wen et al, 2023; Yang et al, 2021). Briefly, mice were placed in a Plexiglas chamber on an elevated mesh screen. Each von Frey filament was applied to the plantar sides of both hind paws 10 times. A quick withdrawal of the paw was regarded as a positive response. The number of positive responses among ten applications was recorded as percentage withdrawal frequency [number of paw withdrawals/10 trials) × 100 = % response frequency].

Paw withdrawal latencies in response to noxious heat stimulation were examined as published (Jia et al, 2022; Mao et al, 2019; Wei et al, 2021; Wen et al, 2023; Yang et al, 2021). Briefly, mice were placed in a Plexiglas chamber on a glass plate. A beam of light emitted from a hole in the light box of a Model 336 Analgesia Meter (IITC Inc. Life Science Instruments, Woodland Hills, CA) was applied to the middle of the plantar surface of each hind paw. A quick lift of the hind paw was regarded as a signal to turn off the light. The length of time between the start and the stop of the light beam was defined as the paw withdrawal latency. For each side, three trials at 5-min intervals were carried out. A cut-off time of 20 s was used to avoid tissue damage to the hind paw.

Paw withdrawal latencies to noxious cold (0 °C) were observed as described (Jia et al, 2022; Mao et al, 2019; Wei et al, 2021; Wen et al, 2023; Yang et al, 2021). Mice were placed in a Plexiglas chamber on the cold aluminum plate, the temperature of which was monitored continuously by a thermometer. The paw withdrawal latency was recorded as the length of time between placement and the first sign of the mouse jumping and/or flinching. Each trial was repeated three times at 10-min intervals on the ipsilateral side. To avoid tissue damage, a cut-off time of 20 s was used.

CPP test was carried out as described with minor modifications (Jia et al, 2022; Mao et al, 2019; Wei et al, 2021; Wen et al, 2023; Yang et al, 2021). Briefly, an apparatus with two Plexiglas chambers connected with an internal door (Med Associates Inc., St. Albans, VT) was used. One of the chambers was made of a rough floor and walls with black and white horizontal stripes, and another one was composed of a smooth floor and walls with black and white vertical stripes. Movement of the mice and time spent in each chamber

were monitored by photobeam detectors installed along the chamber walls and automatically recorded in MED-PC IV CPP software. Mice were first preconditioned for 30 min with full access to two chambers to habituate them to the environment. At the end of the preconditioning phase, basal duration spent in each chamber was recorded within 15 min to check whether animals had a preexisting chamber bias. Mice spending more than 80% or less than 20% of total time in any chamber were excluded from further testing. The conditioning protocol was performed for the following 3 days with the internal door closed. The mice first received an intrathecal injection of saline (5 μl) specifically paired with one conditioning chamber in the morning. Six hours later, lidocaine (0.8% in 5 μl of saline) was given intrathecally paired with the opposite conditioning chamber in the afternoon. Lidocaine at this dosage did not affect motor function. The injection order of saline and lidocaine was switched every day. On the test day, at least 20 h after the conditioning, the mice were placed in one chamber with free access to both chambers. The duration of time that each mouse spent in each chamber was recorded for 15 min. Score differences were calculated as test time-preconditioning time spent in the lidocaine chamber.

Locomotor function was examined as described (Jia et al, 2022; Mao et al, 2019; Wei et al, 2021; Wen et al, 2023; Yang et al, 2021). Three reflexes were conducted as follows. For the placing reflex, the placed positions of the hind limbs were slightly lower than those of the forelimbs, and the dorsal surfaces of the hind paws were brought into contact with the edge of a table. Whether the hind paws were placed on the table surface reflexively was recorded. For the grasping reflex, after the mouse was placed on a wire grid, whether the hind paws grasped the wire on contact was recorded. For the righting reflex, when the mice were placed on its back on a flat surface, whether it immediately assumed the normal upright position was recorded. Each trial was repeated five times at 5-min intervals and the scores for each reflex were recorded based on counts of each normal reflex.

## Cell culture and transfection

Human neuroblastoma cell line (SK-N-BE(2)-C; ATCC, Manassas, VA) cultures and DRG neuronal cultures were prepared as previously described (Jia et al, 2022; Mao et al, 2019; Wei et al, 2021; Wen et al, 2023; Yang et al, 2021). Briefly, SK-N-BE(2)-C cells were cultured in Minimum Essential medium/NEAA: F-12 (Gibco/ThermoFisher Scientific) containing 20% fetal bovine serum (FBS), w/o L-glutamine (200 mM) (Gibco/ThermoFisher Scientific), and 100 units/ml Penicillin and 100 μg/ml Streptomycin (Quality Biological, Gaithersburg, MD). For primary DRG neuronal cultures, after CD1 male mice (≥4 weeks) were euthanized with isoflurane, all DRGs were collected in cold Neurobasal Medium (Gibco/ThermoFisher Scientific) containing 10% fetal bovine serum (JR Scientific, Woodland, CA), 5 mL L-glutamine (200 mM) (Gibco/ThermoFisher Scientific), 10 mL B-27® Supplement (50x) (Gibco/ThermoFisher Scientific), 100 units/ml Penicillin and 100 μg/ml Streptomycin (Quality Biological, Gaithersburg, MD). The DRGs were then treated with enzyme solution (5 mg/ml dispase, 1 mg/ml collagenase type I) in Hanks' balanced salt solution (HBSS) without $Ca^{2+}$ and $Mg^{2+}$ (Gibco/ThermoFisher Scientific). After trituration and centrifugation, dissociated cells

were resuspended in mixed Neurobasal Medium and plated in a six-well plate coated with 50 μg/ml poly-D-lysine (Sigma, St. Louis, MO) at $1.5–4 \times 10^5$ cells. The cell lines or DRG neurons were incubated at 5% $CO_2$ and 37 °C. On the second day, 4–10 μl of virus ($6–9 \times 10^{12}$ tu/μl), 1 μmol of paclitaxel or 100 nM of siRNA (transfected with Lipofectamine 2000) was added to each 2 ml well. The cells were collected 3 days later.

## In situ hybridization histochemistry and co-immunohistochemistry

Mice were deeply anesthetized with isoflurane and transcardially perfused with 25–30 ml of 0.1 M phosphate-buffered saline (PBS, pH 7.4) followed by 30–50 ml of 4% paraformaldehyde in 0.1 M PBS. Following perfusion, DRG was harvested, post-fixed for 4–6 h at 4 °C in the same fixative solution, and cryoprotected in 30% sucrose overnight. 20-μm transverse sections were cut on a cryostat. Every third section was collected from each DRG. In situ hybridization histochemistry (ISHH) was carried out by using a protocol tailored to the IsHyb In Situ Hybridization (ISH) Kit (K2191050; Miramar Beach, FL) with minor modification (Pan et al, 2021). The probe specifically for *Nt3* mRNA was prepared by in vitro transcription and labeled with digoxigenin-dUTP according to the manufacturer's instructions (Roche Diagnostics, Indianapolis, IN). After being treated with proteinase K and pre-hybridized, the sections were hybridized with digoxigenin-dUTP-labeled *Nt3* probes for over two nights at 62 °C. After being blocked for 1 h at RT in 0.01 M PBS/0.3% Triton X-100 containing 4% goat serum, the sections were incubated with AP-conjugated anti-digoxigenin antibody plus chicken anti-β-tubulin III (1:200, catalog number: MAB5564; EMD Millipore; Danvers, MA), rabbit anti-glutamine synthetase (1:500, catalog number: G2781; Sigma-Aldrich; St. Louis, MO), mouse anti-calcitonin gene-related peptide (CGRP, 1:50; catalog number: ab283568; Abcam), rabbit anti-NF200 (1:100, catalog number: N4142; Sigma-Aldrich), biotinylated IB4 (1:100, catalog number: L2140; Sigma-Aldrich) or mouse anti-TH (1:200, catalog number: sc-25269; Santa Cruz; Dallas, TX) at overnight at 4 °C. After being washed, the sections were incubated with avidin labeled with FITC (1:200, Sigma-Aldrich) or goat anti-rabbit antibody, anti-chicken antibody or anti-mouse antibody conjugated with Cy2 (1:200, Jackson ImmunoResearch Labs; West Grove, PA) at room temperature for 1 h. The fluorescent signals were developed with Fast Red. The immuno-fluorescent images were captured under a Leica DMI4000 fluorescence microscope (Leica) with a DFC365 FX camera (Leica). Number of the neurons (with nucleus) double-labeled by *Nt3* mRNA and each marker and number of the neurons (with nucleus) single labeled by each marker in each section were counted. At least 3–4 sections per DRG were examined. The average percentage of *Nt3* mRNA-positive neurons per section within each marker was calculated.

## Western blotting assay

Protein extraction and Western blotting were carried out according to our published protocols (Jia et al, 2022; Mao et al, 2019; Wei et al, 2021; Wen et al, 2023; Yang et al, 2021). Briefly, four DRGs were pooled to obtain enough protein. DRGs or spinal cord were

homogenized and the cultured cells ultrasonicated on ice with the lysis buffer (10 mM Tris, 1 mM phenylmethylsulfonyl fluoride, 5 mM MgCl2, 5 mM EGTA, 1 mM EDTA, 1 mM DTT, 40 µM leupeptin, 250 mM sucrose). After the homogenates were centrifuged at $1000 \times g$ for 15 min at 4 °C, the supernatant (membrane/cytosolic fractions) was collected. The protein concentrations were measured using the Bio-Rad protein assay (Bio-Rad). Twenty µg per sample or 0.05 µg recombinant human NT3 protein (SinoBiological, Wayne, PA) were loaded onto a 4–15% stacking/7.5% separating SDS-polyacrylamide gel (Bio-Rad Laboratories, Hercules, CA). The protein was then electrophoretically transferred onto a polyvinylidene difluoride membrane (Bio-Rad Laboratories). After the membrane was first incubated in the blocking buffer (5% nonfat milk plus 0.1% Tween-20 in the Tris-buffered saline) for 1 h at room temperature, they were incubated with primary antibodies at 4 °C overnight or for two nights. The primary antibodies included rabbit anti-NT3 (1:250; Alomone Labs, catalog number: ANT-003; Jerusalem, Isreael), rabbit anti-MCP-1 (1:1000, catalog number: ab315478; Abcam; Waltham, MA), rabbit anti-p75NTR (1:1000, catalog number: 07-476; Sigma-Aldrich), rabbit anti-TrkC (1:400, catalog number: ANT-020; Alomone), rabbit anti-ERK1/2 (1:1000, catalog number: 4370S; Cell Signaling Technology), rabbit anti-phosphorylated ERK1/2 (p-ERK1/2; 1:1000; catalog number: 3670S; Cell Signaling Technology), mouse anti-GFAP (1:1000, catalog number: 3670S; Cell Signaling Technology), rabbit anti-GAPDH (1:1000, catalog number: G9545; Sigma-Aldrich) and mouse anti-actin (1:1000, catalog number: sc-47778; Santa Cruz). The proteins were detected by horseradish peroxidase–conjugated anti-rabbit and anti-mouse secondary antibodies (1:5000, Jackson ImmunoResearch) at room temperature for 1 h and visualized by western peroxide reagent and luminol/enhancer reagent (Clarity Western ECL Substrate, Bio-Rad) using ChemiDoc XRS System with Image Lab software (Bio-Rad). The intensity of the band was quantified with densitometry using ImageJ software. All bands were normalized to actin or GAPDH.

### Quantitative real-time RT-PCR assay

RNA extraction and quantitative real-time RT-PCR assay were carried out according to our published protocol (Jia et al, 2022; Mao et al, 2019; Wei et al, 2021; Wen et al, 2023; Yang et al, 2021). Briefly, four DRGs were pooled to obtain enough protein. Total RNA was extracted from the DRG by using TRIzol-chloroform methods (Invitrogen), treated with an overdose of deoxyribonuclease I (New England Biolabs), and reverse-transcribed with the ThermoScript Reverse Transcriptase (Invitrogen/ThermoFisher Scientific) and oligo(dT) primers (Invitrogen/ThermoFisher Scientific). The template (4 µl) was amplified in a Bio-Rad CFX96 real-time PCR system by using the primers for *Nt3*, *Ccl2* or *Tuba1a* mRNAs (Table EV2). Each sample was repeated 3 times with a 20-µl reaction volume containing 200 nM forward and reverse primers, 10 µl of SsoAdvanced Universal SYBR Green Supermix (Bio-Rad Laboratories), and 20 ng of cDNA. The PCR amplification included 30 s at 95 °C, 30 s at 60 °C, 30 s at 72 °C, and 5 min at 72 °C for 39 cycles. All PCR data were normalized to an internal control *Tuba1a*. The ratios of mRNA levels were calculated using the ΔCt method ($2 - \Delta\Delta Ct$).

### Single-cell RT-PCR assay

The freshly cultured DRG neurons from adult mice were prepared as described previously (Du et al, 2022; Pan et al, 2021; Wang et al, 2023). Four hours after plating, under an inverted microscope equipped with a micromanipulator and microinjector, a single living large (> 35 µm), medium (25–35 µm), and small (< 25 µm) DRG neurons were collected into a PCR tube containing 6–8 µl of cell lysis buffer (Signosis, Sunnyvale, CA) and centrifuged. The lysis solution was aliquoted into distinct PCR tubes for different genes. The RT-PCR procedure was carried out with the Single-Cell RT-PCR Assay Kit according to the instructions provided by the manufacturer (Signosis). The nest-PCR primers used were listed in Table EV2. The PCR products were analyzed on ethidium bromide-stained 2% agarose gels.

### Plasmid construction and virus production

The full-length sequences of *Nt3* mRNA extracted from mouse DRG were reverse-transcribed using the SuperScript IV One-Step RT-qPCR System with Platinum Taq High Fidelity Kit (Invitrogen/Thermo-Fisher Scientific) and amplified by PCR with gene-specific primers listed in the Table EV2. The resulting segment was inserted into the pro-viral plasmid at the XhoI and NotI restriction sites. The recombinant clones were verified using DNA sequencing. The resulting vectors expressed the genes under the control of the cytomegalovirus promoter. For packaging of AAV5 particles, HEK-293 cells were transfected with the pro-viral plasmid expressing *Nt3* mRNA or *Gfp* using a PEI Transfection method with pHelper. Three days later, the transfected cells were collected and AAV5 particles were purified using the AAV-pro Purification Kit (Takara, Mountain View, CA). The titer was determined by AAV real-time PCR titration kit (Takara).

### Statistical analysis

For in vitro experiments, the cells were evenly suspended and then randomly distributed in each tested well. For in vivo experiments, the animals were distributed into various treated groups randomly. All data were given as means ± SD. Data distribution was assumed to be normal, but this was not formally tested. The data were statistically analyzed with two-tailed, paired/unpaired Student's *t* test and a one-way or two-way ANOVA. When ANOVA showed a significant difference, pairwise comparisons between means were tested by the post hoc Tukey method (SigmaStat, San Jose, CA). Significance was set at $P < 0.05$.

## Data availability

All the data related to this study are available in the manuscript.

The source data of this paper are collected in the following database record: biostudies:S-SCDT-10_1038-S44319-025-00534-1.

## Peer review information

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

## Acknowledgements

This work was supported by grants R01NS111553 and RFNS113881 to YXT and R01NS117484 to HH and YXT from the National Institutes of Health (Bethesda, Maryland, USA).

## Author contributions

**Dilip Sharma**: Conceptualization; Data curation; Validation; Investigation; Writing—original draft; Project administration. **Xiaozhou Feng**: Conceptualization; Data curation; Validation; Investigation; Visualization; Project administration. **Bing Wang**: Conceptualization; Supervision; Investigation; Methodology. **Huijie Shang**: Conceptualization; Data curation; Supervision; Investigation; Methodology. **Bushra Yasin**: Methodology; Writing—review and editing. **Alex Bekker**: Data curation; Writing—review and editing. **Huijuan Hu**: Data curation; Supervision; Writing—review and editing. **Yuan-Xiang Tao**: Conceptualization; Data curation; Supervision; Funding acquisition; Validation; Investigation; Writing—original draft; Project administration; Writing—review and editing.

Source data underlying figure panels in this paper may have individual authorship assigned. Where available, figure panel/source data authorship is listed in the following database record: biostudies:S-SCDT-10_1038-S44319-025-00534-1.

## Disclosure and competing interests statement

The authors declare no competing interests.

# Expanded View Figures

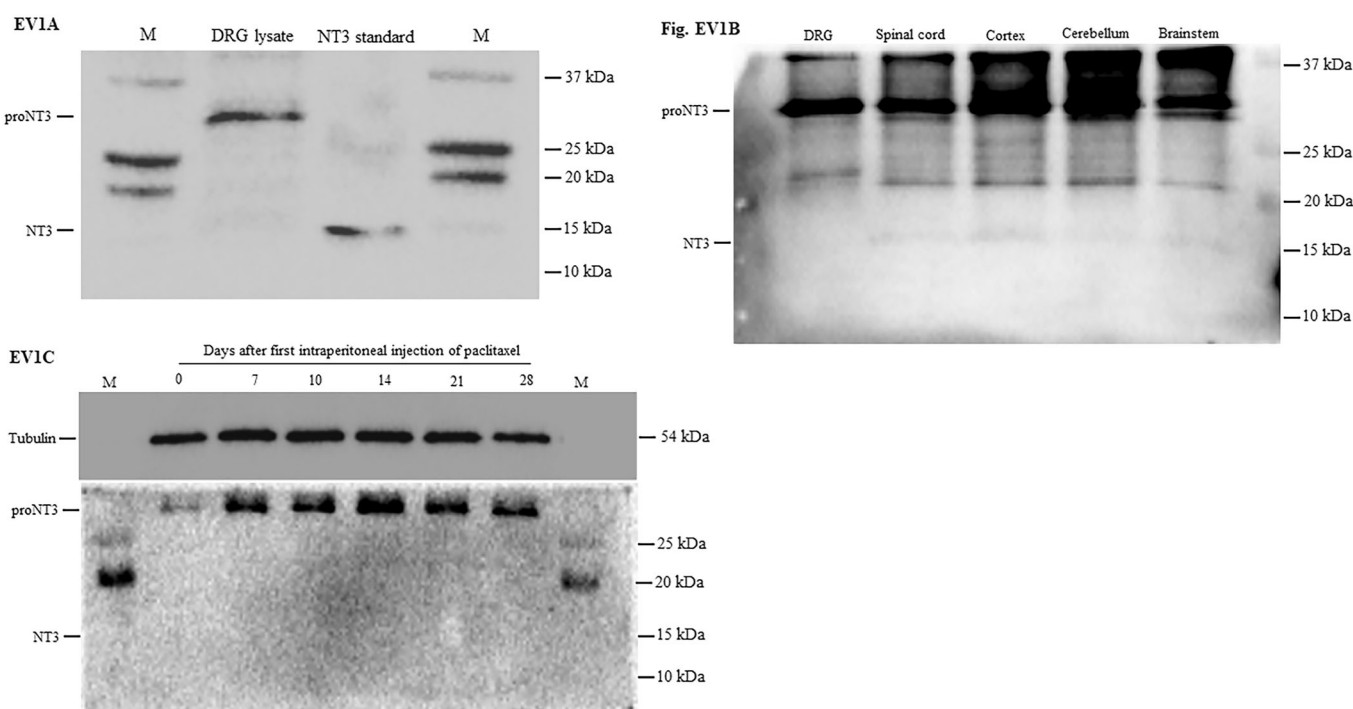

**Figure EV1.   Expression of proNT3, but not NT3, is detected in the DRG of naive and paclitaxel-treated mice.**

(**A**) Total cellular lysates from the DRG of naive mice and recombinant NT3 standard (SinoBiological) are Western blotted with an anti-NT3 antibody (ANT-003, Alomone Labs) to demonstrate that this antibody recognizes both pro-NT3 and NT3. proNT3 (~32 kDa), but not NT3 ( ~ 14.5 kDa), is detected in DRG lysate from naive mice. (**B**) pro-NT3 is highly expressed in the DRG, spinal cord, cortex, cerebellum and brainstem. In contrast, NT3 is undetected in the DRG and expressed weakly in the spinal cord, cortex, cerebellum and brainstem. (**C**) Expression of proNT3 is time-dependently increased in the DRG after first intraperitoneal (i.p.) injection of paclitaxel. NT3 is not detected in the DRG on days 0, 7, 10, 14, 21, and 28 after the first i.p. paclitaxel injection. M: Molecular weight marker. Source data are available online for this figure.

