## [Peer Review File · EMBO Reports]

Proneurotrophin-3 in DRG neurons contributes to chemotherapy-induced neuropathic pain

Dilip Sharma, Xiaozhou Feng, Bing Wang, Huijie Shang, Bushra Yasin, Alex Bekker, Huijuan Hu, and Yuan-Xiang Tao

Corresponding author(s): Yuan-Xiang Tao (yuanxiang.tao@njms.rutgers.edu)

Review Timeline:

Submission Date:	6th Jun 25
Editorial Decision:	30th Jun 25
Revision Received:	30th Jun 25
Accepted:	11th Jul 25

Editor: Esther Schnapp

Transaction Report:

Dear Prof. Tao,

Thank you for the submission of your corrected manuscript. I sent it back to one referee and we received the positive comments below. Can you please address the last few minor suggestions by this referee, and the last editorial notes before we can proceed with the official acceptance of your manuscript. Upon the publication of this corrected ms we will retract the previous, published version. We therefore need all files again for this new ms.

- Please submit again the previous author checklist with this new ms and may be double check whether all answers are still correct.

- All figures should be uploaded as individual, high resolution figure files, in TIFF, EPS or PDF format. The legend needs to be removed from the figure file of Fig EV1 and added to the manuscript text, after the main figure legends, and with the heading "Expanded View Figure Legend".

- Tables EV1 and EV2 should be uploaded as separate files.

- With the current ms, only updated western blot source data (SD) is uploaded - please upload the source data as it was in the previous version of the manuscript (1 SD file per figure) and add the new source data as needed. You can upload the source data for Fig EV1 as a separate file.

Referee #1:

This corrigendum addresses an important correction in the identification of neurotrophin-3 (NT-3) versus its precursor form, proneurotrophin-3 (proNT-3), based on Western blot (WB) analysis. The authors clarify that the primary antibody used recognizes both NT-3 and proNT-3, and the uncropped WB images presented in EV1A-1C convincingly demonstrate that the observed changes reflect alterations in proNT-3, not mature NT-3. The inclusion of recombinant human NT-3 as a positive control further substantiates this interpretation.

While this error could have been recognized in the initial submission, the correction does not undermine the scientific rigor of the study or its overall conclusions, which remain valid and well-supported.

A particularly intriguing finding is the sustained elevation of proNT-3 levels even weeks after nerve injury. Do the authors have any mechanistic insights or hypotheses regarding why proNT-3 remains uncleaved into mature NT-3 for such an extended period post-injury?

Minor Issue:

On page 15, the authors cite studies by Feryhough, Tender, and Wang. Were molecular weights explicitly reported in these studies? If not, it may be premature to conclude that the detected proteins in those reports were indeed proNT-3. Clarifying this point would strengthen the discussion and ensure consistency in interpretation across the literature.

The point-by-point responses to the reviewer's comments:

Response to the Reviewer #1 comments:

A particularly intriguing finding is the sustained elevation of proNT-3 levels even weeks after nerve injury. Do the authors have any mechanistic insights or hypotheses regarding why proNT-3 remains uncleaved into mature NT-3 for such an extended period post-injury?

We appreciate the reviewer's comments. In the revision, we added the following discussion (page 15, lines 3-5): "It is intriguing why proNT3 is not cleaved into mature NT3 in the DRG under normal conditions and up to 21 days post-paclitaxel injection. Whether furin and other proconvertases are absent or present at low levels in the DRG remains to be determined."

Minor Issue:

On page 15, the authors cite studies by Feryhough, Tender, and Wang. Were molecular weights explicitly reported in these studies? If not, it may be premature to conclude that the detected proteins in those reports were indeed proNT-3. Clarifying this point would strengthen the discussion and ensure consistency in interpretation across the literature.

We reviewed these 3 references, in which ELSA and immunohistochemistry for NT3 were used. Given that the molecular weight of NT3 was not reported in these previous studies, we have removed the statement to avoid the misleading information (page 15, lines 5-7).

Prof. Yuan-Xiang Tao
Rutgers university
Anesthesiology
185 S. Orange Ave.
MSB, E-661
Newark, New Jersey 07103
United States

Dear Prof. Tao,

I am very pleased to accept your manuscript for publication in the next available issue of EMBO reports. Thank you for your contribution to our journal.

Your manuscript will be processed for publication by EMBO Press. It will be copy edited and you will receive page proofs prior to publication. Please note that you will be contacted by Springer Nature Author Services to complete licensing and payment information. (I am actually not sure whether you will need to pay publication charges, but you will be informed about the next steps.)
